# The structure of iPLA$_2$β reveals dimeric active sites and suggests mechanisms of regulation and localization

Konstantin R. Malley[1], Olga Koroleva[1], Ian Miller[1], Ruslan Sanishvili[2], Christopher M. Jenkins[3], Richard W. Gross[3,4,5,6] & Sergey Korolev [1]

Calcium-independent phospholipase A$_2$β (iPLA$_2$β) regulates important physiological processes including inflammation, calcium homeostasis and apoptosis. It is genetically linked to neurodegenerative disorders including Parkinson's disease. Despite its known enzymatic activity, the mechanisms underlying iPLA$_2$β-induced pathologic phenotypes remain poorly understood. Here, we present a crystal structure of iPLA$_2$β that significantly revises existing mechanistic models. The catalytic domains form a tight dimer. They are surrounded by ankyrin repeat domains that adopt an outwardly flared orientation, poised to interact with membrane proteins. The closely integrated active sites are positioned for cooperative activation and internal transacylation. The structure and additional solution studies suggest that both catalytic domains can be bound and allosterically inhibited by a single calmodulin. These features suggest mechanisms of iPLA$_2$β cellular localization and activity regulation, providing a basis for inhibitor development. Furthermore, the structure provides a framework to investigate the role of neurodegenerative mutations and the function of iPLA$_2$β in the brain.

[1] Edward A. Doisy Department of Biochemistry and Molecular Biology, Saint Louis University School of Medicine, St. Louis, MO 63104, USA. [2] GM/CA@APS, Advanced Photon Source, Argonne National Laboratory, Argonne, IL 60439, USA. [3] Division of Bioorganic Chemistry and Molecular Pharmacology, Department of Medicine, Washington University School of Medicine, 660 South Euclid Avenue, Campus Box 8020, Saint Louis, MO 63110, USA. [4] Department of Developmental Biology, Washington University School of Medicine, Saint Louis, MO 63110, USA. [5] Department of Medicine, Center for Cardiovascular Research, Washington University School of Medicine, Saint Louis, MO 63110, USA. [6] Department of Chemistry, Washington University, Saint Louis, MO 63130, USA. Correspondence and requests for materials should be addressed to S.K. (email: sergey.korolev@health.slu.edu)

Calcium-independent phospholipase A$_2$β (iPLA$_2$β, also known as PLA2G6A or PNPLA9) hydrolyses membrane phospholipids to produce potent lipid second messengers[1,2]. Due to its emerging role in neurodegeneration and a strong genetic link to Parkinson's disease (PD)[3–9], the gene coding for iPLA$_2$β was designated as *PARK14*. Originally isolated from myocardial tissue as an activity stimulated during ischemia[10,11], the enzyme displays several specific features including calcium-independent activity, a preference for plasmalogen phospholipids with arachidonate at the *sn*-2 position, an interaction with ATP[12] and inhibition by calmodulin (CaM) in the presence of Ca$^{2+}$[13]. It was also isolated from macrophages, where it was thought to act as a housekeeping enzyme, maintaining the homeostasis of the lipid membrane[14]. Subsequent studies using the mechanism-based inhibitor bromoenol lactone (BEL) revealed involvement of the enzyme in (1) agonist-induced arachidonic acid release[15]; (2) insulin secretion[16]; (3) vascular constriction/relaxation by Ca$^{2+}$ signaling through store-operated calcium entry[17,18]; (4) cellular proliferation and migration[19,20]; and (5) autophagy[21,22]. Alterations in iPLA$_2$β function have demonstrated its role in multiple human pathologies including cardiovascular disease[1,23,24], cancer[25–27], diabetes[28,29], muscular dystrophy[30], nonalcoholic steatohepatitis[31], and antiviral responses[32]. Correspondingly, inhibitors of iPLA$_2$β have been sought for therapeutic applications. Highly selective fluoroketone inhibitors have been designed[33–35] and successfully applied in mouse models of diabetes[36] and multiple sclerosis[37]. Recently, numerous mutations have been discovered in patients with neurodegenerative disorders such as infantile neuroaxonal dystrophy (INAD)[38–40] and PD[3–9]. The protein was also found in Lewy bodies and its function was connected to idiopathic PD[22,41].

The mechanisms of iPLA$_2$β function in diverse signaling cascades and its role in disease remain poorly understood. More than half of the iPLA$_2$β amino acid sequence is comprised of putative protein-interaction domains and motifs (Fig. 1a and Supplementary Figure 1). The sequence can be divided into three parts: the N-terminal domain, the ankyrin repeat (AR) domain (ANK) and the catalytic domain (CAT)[42]. The lipid hydrolysis is executed by a Ser-Asp catalytic dyad in close spatial proximity to a glycine-rich motif. The CAT domain is homologous to patatin, a ubiquitous plant lipase[43]. The AR is a 33-residue motif consisting of a helix–turn–helix structure followed by a hairpin-like loop forming a conserved L-shaped structure. ARs are found in thousands of proteins and have evolved as a highly specific protein recognition structural scaffold[44]. In different proteins, 4 to 24 ARs can be stacked side-by-side forming elongated linear structures. Five conserved amino acids form a hydrophobic core holding the helical repeats together. The remaining amino acids are variable, but the three-dimensional structure of the AR is highly conserved.

The cellular localization of iPLA$_2$β is tissue-specific and dynamic (review and references are in[45]). Different variants of iPLA$_2$β are associated with the plasma membrane, mitochondria, endoplasmic reticulum, and the nuclear envelope. iPLA$_2$β lacks transmembrane domains, but is enriched in putative protein-interaction motifs. Those include several proline-rich loops and the extended ANK domain with seven or eight ARs capable of interacting with multiple cognate receptor proteins[44,46]. However, relatively little is known about iPLA$_2$β protein-interaction mechanisms. It binds CaM kinase (CaMKIIβ) in pancreatic islet β-cells[47] and the endoplamic reticulum (ER) chaperone protein calnexin (Cnx)[48]. The functional significance and mechanisms of both interactions remains unknown. Pull-down of proteins isolated from β-cells under mild detergent treatment revealed a number of other proteins from different cellular compartments, including transmembrane proteins[48]. iPLA$_2$β was also found in

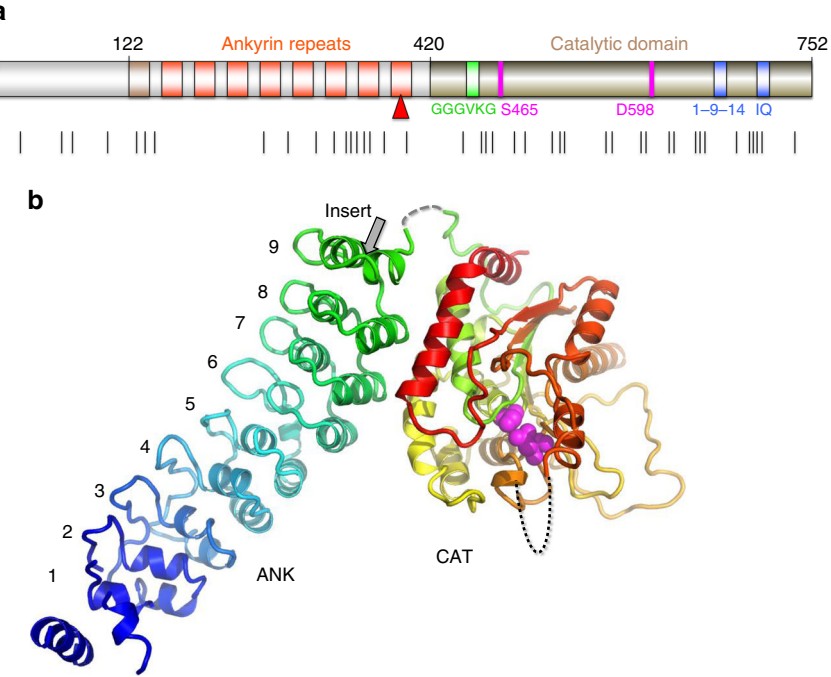

**Fig. 1** Sequence motifs and the structure of iPLA$_2$β. **a** Domain composition of iPLA$_2$β. ARs are shown in orange with the novel AR$_1$ in dark orange, catalytic residues are in magenta, poly-Gly region is in green, and putative CaM-binding motifs in blue. Black lines underneath mark INAD and PD mutations. **b** Cartoon representation of the iPLA$_2$β monomer color-coded in a rainbow scheme with the N terminus in blue and the C terminus in red. The catalytic dyad is shown by magenta spheres. The location of the unstructured loop between ANK and CAT domains is indicated by the dashed gray line and of the disordered membrane-interacting loop by the black dotted line. The position of the proline-rich insert in the long variant is shown by the grey arrow in this panel and by the red triangle in panel a

the Arf1 interactome, which regulates cell morphology[49]. Understanding the mechanisms of the diverse iPLA$_2$β functions requires knowledge of its spatial and temporal localization, which are most likely guided by poorly understood protein–protein interactions. Structural studies of iPLA$_2$β are currently limited to identification of the putative CaM-binding sites[50], molecular modeling, and mapping of the membrane interaction loop using hydrogen/deuterium exchange mass spectrometry[51,52].

Here, we present the crystal structure of a mammalian iPLA$_2$β, which revises previous structural models and reveals several unexpected features critical for regulation of its catalytic activity and localization in cells. The protein forms a stable dimer mediated by CAT domains with both active sites in close proximity, poised to interact cooperatively and to facilitate transacylation and other potential acyl transfer reactions. The structure suggests an allosteric mechanism of inhibition by CaM, where a single CaM molecule interacts with two CAT domains, altering the conformation of the dimerization interface and active sites. Surprisingly, ANK domains in the crystal structure are oriented toward the membrane-binding interface and are ideally positioned to interact with membrane proteins. This finding could explain how iPLA$_2$β differentially localizes within a cell in a tissue-specific manner, which is a long-standing question in the field. The structural data also suggest an ATP-binding site in the AR and outline a potential role for ATP in regulating protein activity. These structural features and structure-based hypotheses will be instrumental in deciphering mechanisms of iPLA$_2$β function in different signaling pathways and their associated diseases. Mapping the location of neurodegenerative mutations onto the dimeric structure will shed light on their effect on protein activity and regulation, improving our understanding of iPLA$_2$β function in the brain.

## Results

**Structure of iPLA$_2$β.** The structure of the short variant of iPLA$_2$β (SH-iPLA$_2$β, 752 amino acids) was solved by a combination of selenomethionine single-wavelength anomalous diffraction (SAD) with molecular replacement (MR) using two different protein models. Those include patatin[43], which has a 32% sequence identity to the CAT domain, and four ARs of the ankyrin-R protein[53], with a 20% sequence identity to four C-terminal ARs of iPLA$_2$β (Supplementary Figure 1). Five additional ARs and several loop regions in CAT were modeled into the electron density map. The sequence assignment was guided by position of 51 selenium peaks and the structure was refined using 3.95 Å resolution data (Supplementary Table 1 and Supplementary Figure 2). Residues 1–80, 95–103, 113–117, 129–145, 405–408, and 652–670 were omitted from the final model. Regions 81–94, 104–112, and 409–416 were modeled as alanines. The short variant lacks a proline-rich loop in the last AR (Fig. 1) and sequence numbering in the paper corresponds to sequence of the SH-iPLA$_2$β. The structure of the monomer is shown in Fig. 1b.

The core secondary-structure elements of the CAT domain are similar to that of patatin with root-mean-square deviation (r.m.s.d.) of 3.1 Å for 186 Cα atoms (Supplementary Figure 3a). Consequently, the fold of the CAT domain also resembles that of cytosolic phospholipase A$_2$α (cPLA$_2$α) catalytic domain[54], but to a significantly lesser extent. The active site is localized inside the globular domain as in the patatin structure. However, in iPLA$_2$β, the catalytic residues are more solvent accessible than in patatin (Supplementary Figure 3b). In the latter, the active site is connected to the surface through two narrow channels (Supplementary Figure 3c) and significant conformational changes are required for phospholipid binding. By contrast, in iPLA$_2$β, the active site cavity is wide open and can accommodate phospholipids with long polyunasaturated fatty acid chains.

The periphery and loop regions differ significantly from those in the patatin structure, with two unique extended proline-rich loops in iPLA$_2$β. A long C-terminal α-helix (α7 in patatin[55]) is kinked in the iPLA$_2$β structure and participates in dimerization (described below).

**Conformation of the ANK domain.** The electron density map reveals nine ARs in the structure of SH-iPLA$_2$β, instead of the previously predicted eight. AR$_1$ is formed by residues 120–147 with a less conserved AR signature sequence motif (Supplementary Figure 1). The outer helix of AR$_1$ is poorly ordered and was omitted from the current model. The C-terminal AR$_9$ is formed by residues 376–402. Gln396, which is substituted by the 54-residue proline-rich insert in the long variant (L-iPLA$_2$β), is located in the short loop connecting two helixes of AR$_9$ (gray arrow in Fig. 1b). The orientation of the entire ANK domain is completely unexpected (Figs. 1b, 2b). It is attached to the CAT domain at the side opposite to the membrane-binding surface and was thought to form an extended structure oriented away from the membrane to participate in oligomerization[56]. In the crystal structure, it wraps around the CAT domain towards the predicted membrane-interacting surface. This is achieved by the extended conformation of an 18 amino-acid-long connecting loop, illustrated in Supplementary Figure 4a. Part of the linker is unresolved due to poor electron density; however, the assignment of the ANK and CAT domains to the same molecule is unambiguous in the crystal packing. The outer helices of AR$_7$ and AR$_8$

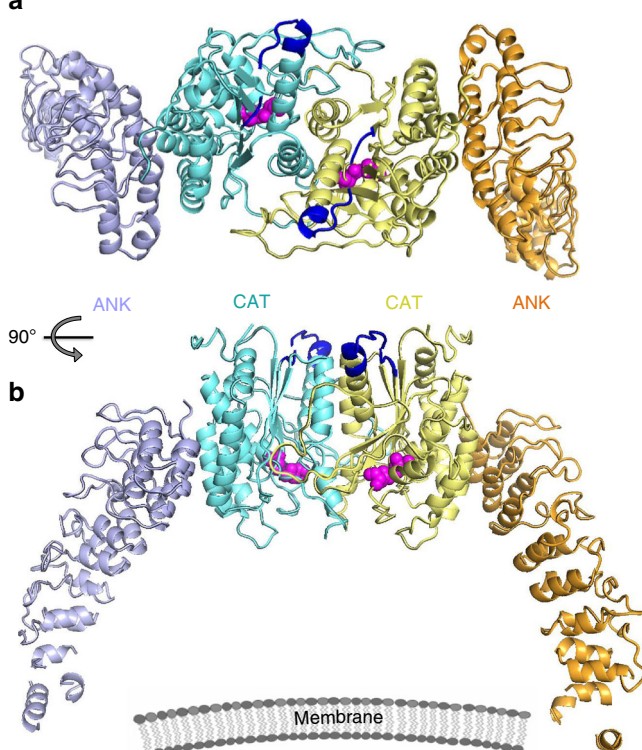

**Fig. 2** Configuration of the iPLA$_2$β dimer in the crystal structure. **a** The CAT and ANK domains of a dimer are shown in cyan and light navy, respectively, in monomer A and in yellow and orange in monomer B. Putative CaM-binding 1-9-14 motifs in both monomers are shown in dark blue. Catalytic dyads are shown by magenta spheres. **b** Same dimer rotated by 90° around horizontal axis. The schematic drawing of a membrane illustrates the orientation of the membrane-binding surface of iPLA$_2$β

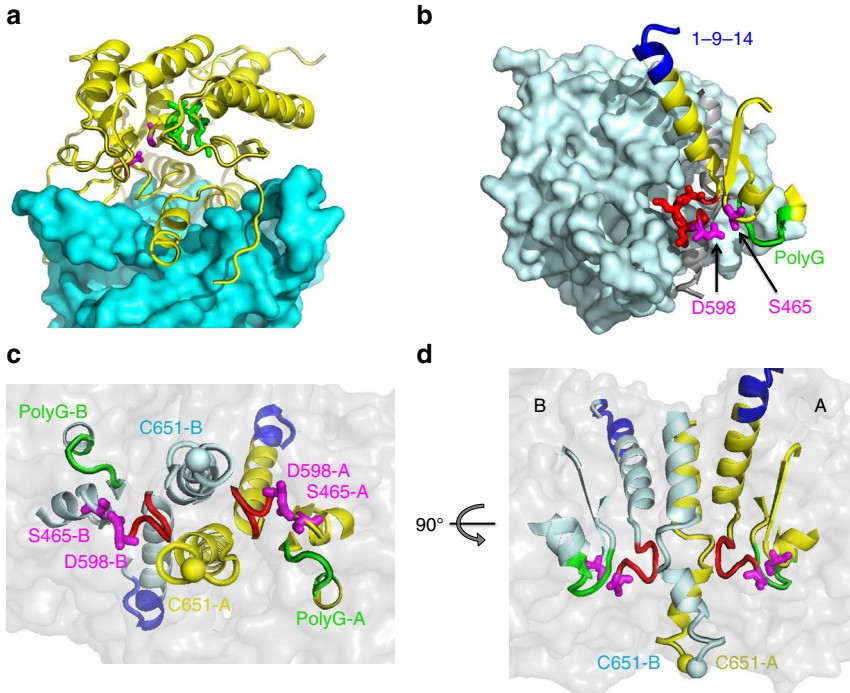

**Fig. 3** Extensive interactions of CAT domains and integrated active sites. **a** Interaction of the CAT domain of molecule B (CAT-B), shown as cyan surface, with the CAT domain of molecule A (CAT-A), shown as yellow cartoon with highlighted catalytic dyad residues (magenta sticks) and the oxyanion hole (green). **b** The proximity of the active site to the dimerization interface is illustrated with surface representation of CAT-B (light cyan) and structural elements of the CAT-A active site shown as yellow cartoon, along with the Ser-Asp dyad of CAT-A (magenta stick representation), the oxyanion hole formed by poly-Gly loop (green), and the π-helix (red) which contains the catalytic Asp. The structured fragment of 1-9-14 motif is shown in blue. **c** The view from the membrane-binding surface of the active sites of a dimer with secondary-structure elements and the individual residues color-coded as in **b** for molecule A and by light cyan for molecule B. A transparent surface of the dimer is shown in grey. C651 residues of the dimer are represented by yellow and light cyan spheres. These cysteines were previously reported to be acylated in the presence of acyl-CoA and are located on the membrane side of the protein surface. **d** Side view of the same structural elements in orientation orthogonal to that in **c**, illustrating the distance of catalytic dyad residues from the membrane-interacting surface and the location of Cys651 at this surface as well as near the dimerization interface

form an extensive hydrophobic interface with CAT. $AR_9$ partially contributes to this interface as well.

**ANK interaction with ATP**. $iPLA_2\beta$ is the only known phospholipase that interacts with ATP[12]. The glycine-rich motif was initially proposed as an ATP-binding site. However, this motif is highly conserved through patatin-like phospholipases, where it forms part of the active site. It is also a common element of α/β hydrolases, where it functions as an oxyanion hole coordinating charge distribution during catalysis[57]. To identify the location of ATP binding in $iPLA_2\beta$, we soaked protein crystals with 2′MeSe-ATP and collected 4.6 Å anomalous data. A single anomalous peak was consistently found near Trp293 of $AR_6$ (Supplementary Figure 5a). An electron density, adjacent to this residue, was also found in the Fo-Fc map calculated from the Se-Met crystal (Supplementary Figure 5b), where ATP was present during protein concentration to improve solubility. This strongly suggests that ATP binds near Trp293. The density was less pronounced in the native crystal, where the nucleotides diffused out of the capillary used for crystallization (see Methods). Since the low resolution of the Se-Met and Se-ATP data did not permit unambiguous modeling of ATP, we did not include it in the final refinement.

Importantly, $AR_6$ (residues 282–308) adopts an unusual conformation. One of its helices is two amino acids shorter than a conventional AR helix. There is a kink of the entire ANK domain at this position, as compared to ankyrin-R. Potential ATP binding at this location, where an elongated ANK domain structure is disrupted by the short α-helix of $AR_6$, suggests that

nucleotides can regulate the conformation and thermodynamic stability of the ANK domain.

To our knowledge, the interaction of ATP with ARs has been reported only once in the literature. TRPV1 binds ATP within the positively charged inner concave surface of three ARs[58]. In $iPLA_2\beta$, $AR_6$ and $AR_7$ also possess several basic residues in a corresponding surface area. However, the position of the anomalous peak next to Trp293 suggests a potential stacking interaction. Interestingly, it was shown that both purine nucleotides, ATP and GTP, have similar effects on $iPLA_2\beta$ activity[12].

**CAT-mediated dimerization of $iPLA_2\beta$**. The crystallographic asymmetric unit is formed by a dimer of $iPLA_2\beta$. In contrast with the original hypothesis of ANK-mediated oligomerization[56], this dimer is formed through CAT domains (Figs. 2, 3). The ANK domains are oriented outwards in opposite directions, forming a ~150-Å-long elongated structure (Fig. 2). Analytical ultracentrifugation (AUC) experiments support the existence of an elongated dimeric structure in solution (Fig. 4a). The experimental molecular weight (MW) was 152 kDa, corresponding to a dimer with a theoretical MW of 170 kDa. The friction ratio of 1.9 (compared to 1.4 for bovine serum albumin (BSA)) corresponds to a significant deviation from a globular shape. The isolated ANK domains did not form oligomers in AUC experiments (Supplementary Figure 6b).

CAT domains interact through an extended, largely hydrophobic dimerization interface with a contact area of ~2,800 Å$^2$ (Fig. 3) formed by three α-helices, several loops, including the

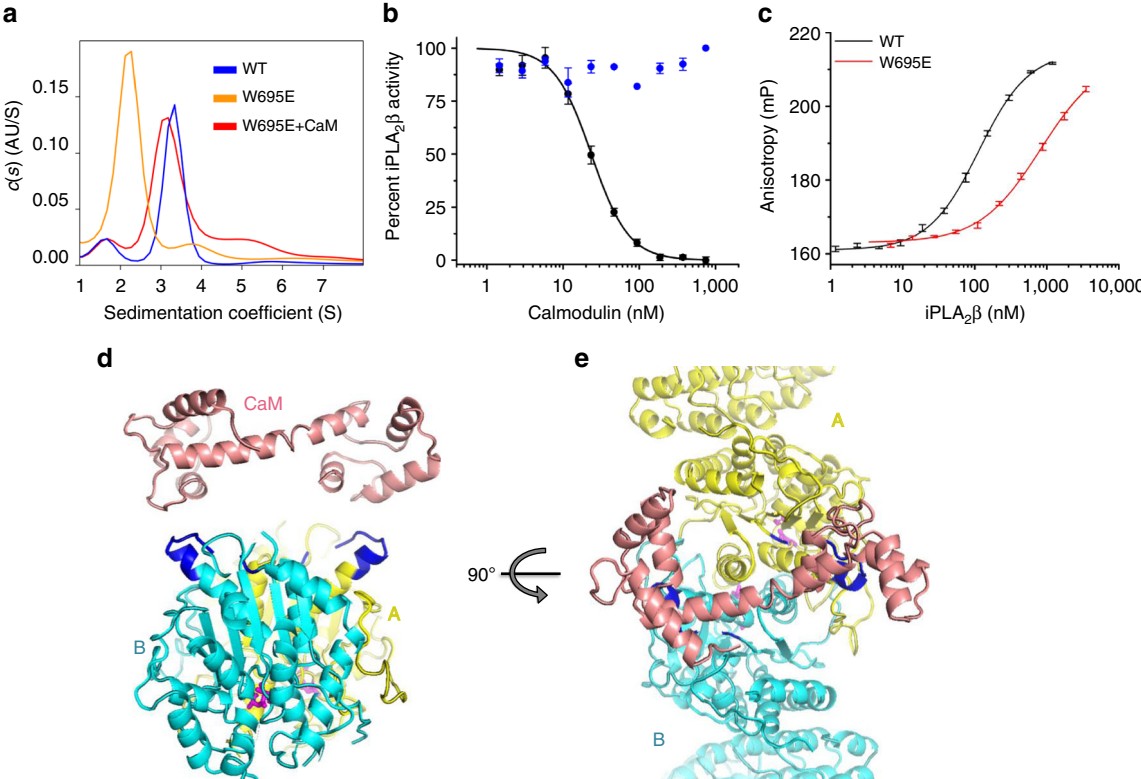

**Fig. 4** iPLA$_2$β dimerization and mechanism of CaM inhibition. **a** Sedimentation velocity s(w) distributions of wild-type iPLA$_2$β shown in blue (peak MW: 152 kDa; theoretical MW: 170 kDa), of the W695E mutant in orange (83 kDa), and of W695E in the presence of CaM/Ca$^{2+}$ in red (138 kDa). Lower MW can be due to unresolved contribution of minor fraction of monomeric species in the latter case. **b** Inhibition of iPLA$_2$β enzymatic activity by CaM in the presence (black) and absence (blue) of Ca$^{2+}$. **c** Interaction of FAM-CaM/Ca$^{2+}$ with iPLA$_2$β as measured by fluorescence anisotropy upon titration by wild-type iPLA$_2$β (black) and the W695E mutant (red). Error bars represent average ± s.e.m of triplicate experiments, which were performed at least twice independently. **d**, **e** Two orthogonal views of the iPLA$_2$β dimer with monomers color coded in cyan and yellow. CaM in the conformation as reported in the 3SJQ PDB structure is placed next to 1-9-14 motifs (highlighted in blue) to illustrate the possibility of CaM interaction with two 1-9-14 motifs of a dimer

long loop 554–570, and a part of the central β-sheet. Such an extensive interaction supports a stable dimer. Correspondingly, iPLA$_2$β dimerizes even at nanomolar concentrations (Supplementary Figure 6a). To probe the dimerization interface, we substituted Trp695 with Glu (W695E). Trp695 forms extensive hydrophobic interactions with the opposite monomer, including a stacking interaction with its counterpart (Supplementary Figure 4b). The mutant is a monomer in solution (Fig. 4a) and is inactive (Supplementary Figure 6d). The W695A mutant exists in equilibrium with both monomeric and dimeric peaks (Supplementary Figure 6c).

The monomers are related by a twofold axis rotational symmetry. Two active centers and the predicted membrane-binding loops[51] are oriented in the same direction (Fig. 3d). Importantly, the active sites are in the immediate vicinity of the dimerization interface and in close spatial proximity to each other (Fig. 3). The catalytic Asp598 is at the beginning of a π-helical loop (599–603) and two leucines of this loop form contacts with the long α-helix (604–624) of the opposite monomer. This arrangement suggests a strong allosteric association between the two active sites and dependence of the catalytic activity on a dimer conformation.

**CaM-binding mechanism.** CaM inhibits iPLA$_2$β enzymatic activity in the presence of calcium. It was proposed to tightly interact with iPLA$_2$β even at low calcium concentrations[59] and to be displaced by active mechanisms, such as covalent modification

of the active site by acyl-CoA[60] or by interaction with a calcium influx factor released from the ER during calcium depletion[61]. Two putative CaM-binding peptides containing the canonical IQ and 1-9-14 motifs were previously isolated by tryptic footprinting and affinity chromatography using CaM-agarose[50]. We measured the $K_i$ of iPLA$_2$β inhibition by CaM using a fluorogenic activity assay with Pyrene-PC (1-hexadecanoyl-2-(1-pyrenedecanoyl)-*sn*-glycero-3-phosphocholine) fluorescent phospholipid liposomes (Supplementary Figure 7a–e). The results revealed a tight calcium-dependent interaction with CaM with a $K_i$ of 23 ± 1.5 nM (s.e.m of three replicates) (Fig. 4b) and a Hill coefficient $n = 2.2 \pm 0.2$, indicating potential cooperativity. Next, we measured the direct interaction of CaM with iPLA$_2$β using fluorescent polarization with fluorescein (FAM)-labeled CaM (Fig. 4c). The dissociation constant of the interaction of CaM with iPLA$_2$β ($K_d$) of 112 ± 5 nM was higher than the $K_i$ measured with unmodified CaM; however, it corresponds to the $K_i$ of FAM-CaM (Supplementary Figure 7g). No cooperativity was observed in the direct binding experiment.

Remarkably, the interaction of FAM-CaM with the monomeric W695E mutant was at least an order of magnitude weaker, with a $K_d > 1,400$ nM (Fig. 4c), suggesting that iPLA$_2$β dimerization is crucial for CaM binding. The interaction of CaM with synthetic isolated FAM-labeled peptides corresponding to 1-9-14 and IQ motifs was even weaker. Affinity towards the 1-9-14 motif ($K_d = 2,500 \pm 400$ nM) was comparable to that of the monomeric W695E mutant. Binding of the IQ motif was significantly weaker ($K_d = 5,900 \pm 800$ nM) (Supplementary Figure 7h). Finally, an

excess of CaM enabled W695E dimerization in sedimentation velocity AUC experiments (Fig. 4a). These data strongly support the model where a single CaM molecule interacts with an iPLA$_2$β dimer and explains the potential cooperativity in the inhibition assay. Furthermore, the two 1-9-14 motifs are located on the same side of the dimer and are ~30 Å apart from each other (Fig. 4d, e). In the structure of the small conductance potassium channel complex with CaM (PDBID: 3SIQ)[62], a single CaM molecule in an extended conformation interacts with the channel dimer and the distance between CaM-binding helixes is also 30 Å. In Fig. 4d, e, CaM from the 3SIQ complex is placed next to an iPLA$_2$β dimer to illustrate comparable distances. At the same time, the conformation of the IQ motif in the tertiary structure makes it an unlikely target of CaM binding. This motif overlaps with a β-strand of the conserved structural core of the molecule and is inaccessible for binding without protein unfolding. Moreover, mutation of the most conserved hydrophobic Ile to a charged Asp (I701D) in the IQ motif did not affect iPLA$_2$β inhibition by CaM (Supplementary Figure 7f). Together, results from solution studies and the conformation of potential CaM-binding sites in the iPLA$_2$β dimer suggest that one CaM molecule interacts with two monomers of the iPLA$_2$β dimer, most likely through the 1-9-14 motifs.

## Discussion

The crystal structure of iPLA$_2$β has revealed several unexpected features underlying its enzymatic activity and mechanisms of regulation. Previous computer modeling studies, based on the patatin structure, proposed an interfacial activation mechanism whereby interaction with membrane leads to opening of a closed active site[34]. In the iPLA$_2$β crystal structure, the active site adopts an open conformation in the absence of membrane interaction (Supplementary Figure 3b). Both active sites of the dimer are wide open and provide sufficient space for phospholipids to access the catalytic centers. This is in contrast to patatin, where only two narrow channels connect the catalytic dyad with the solvent-exposed surface, and conformational changes are required for substrate to access the active site (Supplementary Figure 3c). An open conformation of the active site explains the ability of iPLA$_2$β to efficiently hydrolyze monomeric substrates[13] and the lack of a strong interfacial activation such as observed with cPLA$_2$, where membrane binding increases activity by several orders of magnitude[63].

The dimer is formed by CAT domains tightly interacting through an extensive interface, while ANK domains are oriented outwards from the catalytic core. The existence of the dimer in solution was confirmed by quantitative sedimentation velocity and cross-linking experiments. This configuration was verified by mutagenesis of the observed dimerization interface and a lack of oligomerization by isolated ANK domains. The elongated shape of the dimer contributes to an overestimation of the previously reported oligomeric state in gel filtration analysis due to faster migration of elongated molecules through the size-exclusion matrix. A remote iPLA$_2$β homolog from *Caenorhabditis elegans* also forms a dimer in solution[22].

The catalytic centers are in immediate proximity to the dimerization interface and the activity is likely to depend on the conformation of the dimer. Disruption of the dimer by the W695E mutation yields an inactive enzyme. The active sites are also in close proximity to each other and allosterically connected. Concerted activation of closely integrated active sites should promote rapid responses upon stimulation by ligands, rendering the enzyme an efficient sensor of external perturbations. The close proximity of the active sites provides a plausible explanation of the previously reported activation mechanism

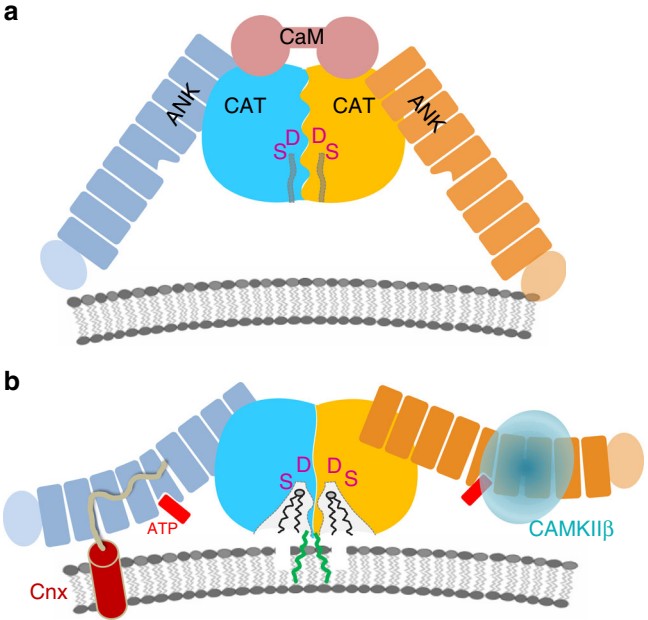

**Fig. 5** The proposed mechanism of iPLA$_2$β regulation and macromolecular interactions. **a** Schematic representation of the iPLA$_2$β dimer in a hypothetical inhibited state bound to CaM. CAT domains are shown in blue and yellow, ANK domains in navy and orange, and a single CaM molecule is represented by two connected circles in pink. Active site cavities are represented by narrow channels (gray lines) leading from the solvent-exposed surface to the Ser/Asp catalytic dyad depicted by magenta. **b** An active conformation of the dimer. CaM dissociation leads to the opening of the active sites. ANK domains are available for interactions with protein partners as illustrated for CAMKIIβ (light cyan transparent sphere), known to interact with ANK domain, and with transmembrane Cnx (shown as transmembrane helix with the C-terminal cytosolic peptide in pale yellow), which could recruit iPLA$_2$β to the membrane. The Cnx-binding site of iPLA$_2$β is not known and the hypothetical interaction with ANK domain is based on similar interaction of AnkB and sodium channel peptide. ATP binding (red) in the middle of the ANK domain could trigger additional conformational changes of the AR. Acylation of C651 by oleoyl-coa (green) can facilitate interaction with the membrane and/or opening of active site channels. Other conformational states are feasible as well, such as CaM-bound inhibited protein at the membrane or an open conformation of active sites in CaM-free form in cytosol, corresponding to the crystallized form

through autoacylation of Cys651. The reaction occurs in the presence of oleoyl-CoA and the modified enzyme is active even in the presence of CaM/Ca$^{2+}$[60]. Cys651 is located at the entrance to the active site at the base of the membrane-binding loop as well as at the dimerization interface (Fig. 3d). Covalent attachment of a long fatty acid chain at this position should increase protein affinity to the membrane and can alter the conformation of a CaM-bound dimer. The close proximity of two active sites provides an explanation for this autoacylation phenomenon important for iPLA$_2$β activation in the heart during ischemia.

An intimate allosteric connection of active sites and the dimerization interface also provides a conceivable mechanism for inhibition by CaM. Indeed, solution studies and location of the putative CaM-binding site strongly suggest that a single CaM binds two molecules of the dimer. We hypothesize that such interactions will lead to conformational changes in the dimerization interface and alter the conformation of both active sites.

A hypothetical model of two potential states of iPLA$_2$β with CaM-bound inactive and CaM-free active dimers is illustrated in Fig. 5. In both states, the enzyme is a dimer. The conformation of the dimerization interface differs in the two states depending on

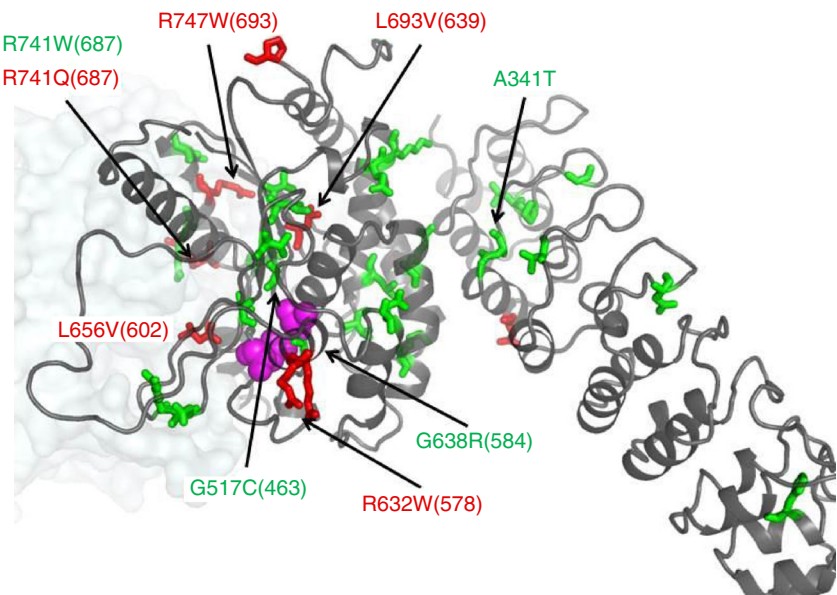

**Fig. 6** Positions of selected INAD and PD mutations. Residues mutated in INAD and PD patients are shown as green and red sticks, respectively, on the cartoon of the iPLA$_2\beta$ monomer. Position of the second monomer is shown as a pale transparent surface. Active site Ser/Asp residues are shown by magenta spheres. Mutations previously tested for enzymatic activity are labeled. Numbers correspond to the human long variant (L-iPLA$_2\beta$) sequence with corresponding numbers of the short variant structure in parentheses

interaction of CaM with the 1-9-14 motif. Allosterically, CaM binding stabilizes a closed conformation of the active sites, which remain open in the absence of CaM. The positions of ATP and of acyl modification are shown in the active form. However, the exact mechanism of activation through autoacylation and the effect of ATP binding on protein activity remain to be further investigated. ANK domains are likely to move out of the conformation observed in crystal structure upon approaching the membrane. In crystals the dimer is shaped as an arch standing on legs formed by the ANK domains (Fig. 2b), with the CAT domains at the top and with their active sites facing downward. The inner radius of the arch is ~80–100 Å. Therefore, in this conformation the ANK domains can prevent the membrane surfaces of larger radii from accessing the catalytic domains. However, the non-specific hydrophobic interactions permit rotational flexibility of interacting domains. Therefore, ANK domains can rotate out of this inhibitory position, while maintaining hydrophobic contacts with the CAT domain. In fact, the relative orientation of the ANK and CAT domains is slightly different between the two monomers of the same crystal. Upon superposition of CAT domains of two monomers, the resulting orientation of ANK domains differs with N-terminal ends shifted by ~12 Å (Supplementary Figure 4c). Similar variation of the ANK domain orientation is a major source of non-isomorphism between different crystals.

The model also illustrates a hypothetical interaction of ANK domains with cytosolic CAMKIIβ and with cytosolic C-terminus of transmembrane Cnx. The iPLA$_2\beta$ field currently lacks a coherent explanation of how the protein localizes to the membrane of various organelles in different cell types and tissues. Elongated ARs form highly specific docking surfaces for different protein partners and 4–8 ARs are capable of binding several proteins. The finding that the ANK domains are oriented towards the membrane-facing side of iPLA$_2\beta$ suggests that ARs represent the most logical site of putative interaction with membrane proteins. Indeed, specific protein recognition is the only known function of ARs. The canonical ARs tether cytoskeletal proteins to ion channels[53].

CAMKIIβ has been reported to interact with the N-terminal part of iPLA$_2\beta$. The function of such an interaction remains poorly understood. It can either modulate the activity of either protein or mediate recruitment of iPLA$_2\beta$ to the membrane since CaMKIIβ forms complexes with actin, an L-type Ca$^{2+}$ channel regulatory protein a1C, and synaptic proteins like NMDA receptors, densin-180 (a *trans*-synaptic protein), and α-actinin (an actin-binding protein). In the case of strong interaction, CaMKIIβ may also play a role of iPLA$_2\beta$ storage pool due to the high abundance of CaMKIIβ in neurons (2% of total protein in hippocampus).

The Cnx-binding domain of iPLA$_2\beta$ remains unknown. This interaction is particularly interesting in light of growing number of data implicating iPLA$_2\beta$ function in ER stress response in β-cells and neurons. Cnx is an ER chaperone protein. It consists of the luminal domain, single transmembrane helix, and a 90 amino-acid-long C-terminal cytosolic tail, which may potentially interact with iPLA$_2\beta$. Interestingly, the interaction of elongated unstructured peptides was previously reported for the AnkB protein with both an autoinhibitory peptide and a peptide of the Nav1.2 voltage-gated sodium channel[64]. Hypothetically, the ANK domain of iPLA$_2\beta$ could similarly interact with a portion of Cnx C-terminal peptide.

The proline-rich 54-residue insert in the long variant is predicted to form an unstructured loop protruding away from AR$_9$, which can also interact with other proteins. Alternatively, it can disrupt the conformation of AR$_9$ and alter orientation of the ANK domain. The hydrophobic interface between ANK and CAT domains and the long flexible linker can allow for significant movement of the ANK domain.

Mutations associated with neurodegeneration are found in all domains, and therefore can affect the enzymatic activity and its regulation as well as macromolecular interactions of iPLA$_2\beta$. In 2006, INAD was linked to mutations in the iPLA$_2\beta$ gene (PARK14)[38], which was later connected to a spectrum of neurodegenerative disorders, correspondingly termed PLAN (recent summary and references in[65]). Those include INAD (INAD1/ NBIA2A), atypical NAD, and idiopathic neurodegeneration with

brain iron accumulation including Karak syndrome (NBIA2B). A different set of mutations was linked to a rapidly progressive young-adult onset dystonia-Parkinsonism [3,5,8,9,66-68]. As shown in Figs. 1a and 6, mutations are spread throughout all domains. Several tested PARK14 mutants retain full[22,69] or partial activity[3], while several tested INAD mutations lead to catalytically inactive enzyme[69]. An interesting example of sensitive allosteric regulation is Arg 741 (corresponding number in SH-iPLA$_2$β is 687) located at the dimerization interface, which is mutated to Trp in INAD, leading to an inactive enzyme, and to Gln in PD with the activity retained. While an Arg to Trp mutation can significantly alter the conformation of the dimerization interface important for catalytic activity, it is unclear what effect a minor Arg to Gln mutation will have and why it causes a late onset (comparatively to INAD) disease. Surprisingly, the A341T mutation in the ANK domain was found to be inactive[69]. This residue is at the ANK/CAT interface and can affect the interactions and stability of the protein. It should be noted that there are very few enzymatic and biochemical studies of the protein and mutants, mostly limited to semi-quantitative measurements. The structure will facilitate in-depth analysis of known mutants and their effect on biochemical properties. This will lead to a better understanding of protein function and the mechanism of activity and regulation in numerous cellular pathways and disease states.

The structure should also facilitate ongoing design of small molecule modulators of iPLA$_2$β for therapeutic purposes. Combined with the analysis of disease-associated mutations, our results clearly demonstrate the importance of N-terminal and ANK domains as well as of peripheral regions of the CAT domain, such as the dimerization interface, for the catalytic activity and its regulation. Together with further knowledge of iPLA$_2$β-binding partners, such allosteric regions can be targets for small molecule binding to inhibit either enzymatic functions or signaling in downstream pathways.

## Methods
**Protein purification**. The pFastBac vector containing the iPLA$_2$β gene cloned from CHO cells with a C-terminal 6XHisTag was used for protein expression[13]. The CHO iPLA$_2$β protein was expressed in Sf9 cells (Invitrogen) using the Bac-to-Bac system. Bacmid DNA was transfected into Sf9 cells with Trans-IT transfection reagent (Mirus Bio). After 4 days, the media were collected as the p0 viral stock. This stock was amplified by adding 1 ml of p0 to 100 ml of $2 \times 10^6$ cells/ml for 96 h, creating the p1 viral stock. Twenty-five milliliters of the amplified p1 was used to infect 500 ml shaker flasks of $2 \times 10^6$ cells/ml for 60 h. The cell pellet was washed with cold phosphate-buffered saline (PBS) and suspended in purification buffer (25 mM HEPES, pH 7.5, 20% glycerol, 0.5 M NaCl, 1 mM TCEP) containing 50 μg/ml each of leupeptin and aprotinin. The cell suspension was frozen in liquid nitrogen and lysed by thawing and sonication at 50% power, 50% duty cycle four times for 2 min each. The lysate was cleared by ultracentrifugation at 100,000 x g for 1 h. Urea of 0.5 M and TCEP of 1 mM were added to the supernatant and mixed with 5 ml of TALON cobalt resin (Clontech) to bind for 1 h at 4 °C. The resin was centrifuged at 800 x g for 1 min to remove the flow-through fraction in batch mode. The resin containing the bound protein was then applied to an empty column, washed sequentially with purification buffer containing 10 mM imidazole (100 ml), 40 mM imidazole (40 ml), and eluted with 15 ml purification buffer containing 250 mM imidazole. iPLA$_2$β and all mutants were >98% pure as determined by sodium dodecyl sulfate-polyacrylamide gel electrophoresis (SDS-PAGE) and Coomassie staining.

The CaM expression plasmid was a gift from M. Shea (University of Iowa). CaM and its mutants were expressed in *E. coli* BL21 star cells (Thermo Fisher) and purified per their detailed protocol[70].

**Crystallization**. iPLA$_2$β was concentrated to 6–8 mg/ml in 10 mM HEPES, pH 7.5, 500 mM NaCl, 10% glycerol, 5 mM ATP, and 1 mM TCEP. Initial crystallization trials were conducted in sitting-drop plates with a Phoenix robot (Art Robbins Instruments) using multiple commercial screens from Hampton Research and Molecular Dimensions. iPLA$_2$β forms crystals within 24 h in several conditions, and after extensive optimization, two primary conditions were selected: 0.1 M bis-tris, pH 5.5, 10% PEG3350, 0.2 M Na/K tartrate, and 0.1 M bis-tris, pH 5.5, 10% PEG3350, 0.2 M sodium acetate. Crystals in sitting-drop conditions displayed poor diffraction (5–7 Å), high X-ray sensitivity and quick deterioration of diffraction power after few days, even while continuing to grow in size. Alternatively, a higher

concentration of protein solution was obtained in the presence of CaM. An equimolar amount of purified CaM was mixed with iPLA$_2$β, reduced with 5 mM dithiothreitol (DTT), and dialyzed in 10 mM HEPES, pH 7.5, 150 mM NaCl, 10% glycerol, 1 mM CaCl$_2$, and 2 mM ATP was added and the proteins were concentrated to 10–12 mg/ml. However, the crystals obtained from iPLA$_2$β in the presence of CaM were identical to those obtained without CaM and SDS-PAGE analysis demonstrated the absence of CaM in the crystals.

Growth of suitable protein crystals (diffracting to better than 4 Å resolution) was enabled by the counter-diffusion method in capillaries, originally using the Granada Crystallization Box (Hampton Research), and, later, using a modified capillary method. In this method, the precipitant solution is layered over a plug of 1% agarose into which a ~7 cm long 1 mm diameter glass capillary tube pre-filled with protein solution is inserted and sealed at the opposite end. The precipitant diffuses through the agarose and gradually mixes with protein throughout the capillary. Importantly, counter-diffusion crystals grew over 1–2 weeks and retained diffraction for up to 2 months. Crystals were harvested from drops or capillaries into a cryoprotectant solution containing 68% mother liquor, 10% PEG3350, 10% ethylene glycol, 10% glycerol, 2% ethanol, and cryo-cooled in liquid nitrogen. Se-Met-labeled iPLA$_2$β was produced in Sf9 cells with the same procedure as native protein, except for using methionine-deficient medium (Expression Systems, Davis, CA, USA) and supplementing with 100 mg/L L-selenomethionine 16 h after infection. The I701D mutant, which had 2–3 times greater expression than wild type, was used for the production of the Se-Met protein. Purification and crystallization of the Se-Met derivative was the same as for native protein.

**Structure determination**. X-ray diffraction data were collected on GM/CA@APS beamlines 23-ID-B and 23-ID-D at the Advanced Photon Source, Argonne National Laboratory. Data collection and refinement statistics are shown in Supplementary Table 1. To identify parts of crystals suitable for data collection, more than 400 samples were tested with the raster method using a small (5–20 μm) beam. The best data sets were collected from elongated crystals using the helical method in order to spread the absorbed dose over larger volume of the crystal and thus reduce the radiation damage to the samples[71]. Data were processed and scaled with HKL2000[72]. It was important to use the "autocorrection" option during scaling, which resulted in rejection of 14% weak reflections due to strong anisotropy of diffraction. Note, that the final statistics after rejection with "autocorrection" was not reported in HKL2000 output. The data were analyzed with Xtriage program within the Phenix program suite and completeness and anisotropy parameters are shown in Supplementary Table 1. Analysis of all data scaled without "autocorrection" in STARANISO server (Global Phasing Limited) resulted in similar number of rejections and anisotropic diffraction limits. Overall, the anisotropy correction reduced the data set completeness, while yielding strong data at resolution higher than 4.4 Å along $c^*$-axis (e.g., $I\sigma(I)$ is 4.5 in highest resolution shell (3.95–4.0 Å), instead of 1.6 for all data reported by HKL2000). Scaling with "autocorrection" also resulted in data with a lower Wilson B factor and, most importantly, in significantly more detailed electron density maps, even comparatively to data processed with other anisotropy reduction programs.

Data from the Se-Met protein crystal were collected at the selenium absorption peak and inflection wavelengths using helical mode and inverse beam geometry with a 30° wedges. Analysis of MAD data at two wavelengths with the Phenix suite[73] did not yield a solution. SAD data using peak wavelength produced a solution with seven selenium peaks. An MR solution was obtained using two different protein models, a patatin[55] and four ARs of an ankyrin-R protein[53]. The structure of patatin was manually trimmed to retain only structural core elements overlapping with CAT domain residues accordingly to the sequence alignment. The Sculptor program within Phenix was used to prepare four ARs from PDB 1N11. The MR solution contained two copies of each domain. Combination of SAD with MR solution resulted in 51 selenium peaks and a high-quality electron density map (Supplementary Figure 2) sufficient for modeling of five additional ARs and several loop regions within the CAT domain. Connectivity of CAT and ANK domains was verified by analysis of all pairs of symmetry-related ANK and CAT domains in the crystal lattice which yielded only one pair with sufficiently short distance. The large number of methionines spread throughout the entire sequence permitted an unambiguous assignment of amino acids. During consecutive steps of structural modeling, combined MR/SAD electron density maps were calculated with one of the domains omitted to avoid model bias. Only one copy of each domain was modeled and the structure was refined using a global NCS function and secondary-structure geometry restriction. After completion of model building, the structure was subsequently refined using 3.95 Å resolution data from the native protein crystal. Simulated annealing composite omit maps were extensively used in model building. Several rounds of Rosetta refinement in Phenix were used for the final model. Phi-psi values of 82% of the residues in the final model are in a favorable region of the Ramachandran plot with 0.4% in an unfavorable conformation. The latter were in loop regions with poor electron density. Residues 1–80, 95–103, 113–117, 129–145, 405–408, and 652–670 were omitted from final model and regions 81–94, 104–112 (numbering in both regions is based on secondary-structure prediction), and 409–416 were modeled as alanine residues.

The 4.6 Å resolution SAD data were collected at selenium peak wavelengths from protein crystals soaked with 2′MeSe-ATP (Jena Bioscience). Combined MR/SAD analysis revealed a single peak. Several alternative models with different

omitted domains or domain fragments were used to avoid model bias. All calculations resulted in an identical position of the selenium peak.

**Fluorescent phospholipase activity assay**. The continuous activity assay was adapted from a protocol used for sPLA₂[74]. Pyrene-PC (Thermo Fisher #H361) (Supplementary Figure 7a, b) was dissolved as a 1 mM stock in dimethyl sulfoxide. The solution was injected into a glass vial containing assay buffer (25 mM HEPES 7.5, 150 mM NaCl, 10% glycerol) over 1 min with shaking to create the substrate mixture. This method resulted in liposomes averaging 100 nm in diameter as determined by dynamic light scattering. One hundred microliters of substrate mixture was added to a black 96-well microplate with a non-binding surface (Corning #3650). Fatty acid-free BSA of 0.2% in the buffer acted as an acceptor for the hydrolyzed 1-pyrenedecanoic acid. Proteins were dialyzed against the assay buffer. iPLA₂β was incubated with different concentrations of CaM with or without 1 mM CaCl₂ for 15 min. The baseline fluorescence of the substrate was recorded for 3 min at 340 nm excitation/400 nm emission using the monochromator of a Biotek Synergy 4 plate reader. Ten microliters of the protein mixture was added to initiate reaction. After a 5 s mixing step, the fluorescence was read every 30 s for 1 h or until the signal reached a plateau (Supplementary Figure 7c). The linear slope of the first 5 min of the reaction was used as the initial velocity. The CaM inhibition data were fit to the Hill equation using Origin 8.6 software. The velocity in fluorescence units per time was quantified in moles using a standard curve of the 1-pyrenedecanoic acid product.

**Fluorescence anisotropy-binding assays**. As CaM has no native cysteine residues, a mutant was engineered at Thr34, as described previously[75], to enable coupling of FAM fluorophore in a site-directed manner. This enabled to measure direct binding of FAM-CaM using fluorescence anisotropy method. The CaM T34C mutant was created by mutagenesis, confirmed by sequencing, and purified with the same procedure as described for the native protein. The labeled protein was separated from excess FAM with phenyl sepharose in the same procedure as for purification. The concentration of labeled protein was measured at 495 nm with a molar extinction coefficient of 68,000/M/cm. For the fluorescence-binding assay, proteins were dialyzed to the assay buffer (25 mM HEPES 7.5, 150 mM NaCl, 10% glycerol). CaM-FAM (30 nM final concentration) was incubated with a series of iPLA₂β concentrations obtained by twofold serial dilution in a 384-well non-binding plate (Corning #3573) in a total volume of 80 μL. After 15 min incubation at 25 °C, the overall fluorescence intensity and the parallel and perpendicular components were read on a Biotek Synergy 4 with 485 nm excitation and 528 nm emission filters. The fluorescence anisotropy was calculated by the Biotek Gen5 software using the following equation: $A = (F_{||} - F_{\perp})/(F_{||} + 2F_{\perp})$, where $F_{||}$ and $F_{\perp}$ are the parallel and perpendicular intensities, respectively. Each experiment was conducted in triplicate at least two independent times and values shown are the average ± s.e.m.

**Analytical ultracentrifugation**. Proteins were extensively dialyzed against AUC buffer (25 mM HEPES 7.5, 500 mM NaCl, 10% glycerol). Sedimentation velocity studies were performed in a Beckman XL-A analytical ultracentrifuge at 20 °C and 35,000 rpm. The absorbance at 280 nm was collected every 4 min for a total of 200 scans. The buffer viscosity and density as calculated by Sednterp (http://www.rasmb.org/sednterp) were 1.04913ρ and 0.01436η, respectively. These values were used to fit the data to the Lamm equation in SEDFIT software[76] using the continuous c(s) distribution model. Graphs were prepared using GUSSI software (UT Southwestern).

**Data availability**. Atomic coordinates and structure factors for the iPLA₂β structure have been deposited in the Protein Data Bank under accession code PDBID 6AUN. All reagents and relevant data are available from the authors upon request.

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

## Acknowledgements

We thank Aaron Naatz for help in purifying of *E. coli* expression constructs, Praveen Subramanian for preliminary work on fluorescence assays of PLA₂ enzymes, and all members of the Korolev lab for helpful discussions. We are grateful to Nicola Pozzi, Enrico Di Cera, David Ford, Jane McHowat, and William S. Sly for extremely helpful discussions and to Joel Eissenberg for manuscript preparation. GM/CA-CAT beamlines 23-ID-B and 23-ID-D at the Advanced Photon Source, Argonne National Laboratory are funded in whole or in part from the NCI (ACB-12002) and the NIGMS (AGM-12006). This research was supported by the American Heart Association Grant-in-Aid #0665513Z, Center for Advancement of Science in Space (CASIS) grant CASIS-2012-1, NIH/NINDS grant R21NS094854 (to S.K.) and NIH/NHLBI grant R01HL118639 (to R. W.G.).

## Author contributions

K.R.M. and S.K. designed, performed, and analyzed experiments, including crystallographic data collection and refinement. C.M.J. and R.W.G. provided purified protein for initial crystallization experiments, plasmid for protein production, and provided expertise in studying the effectors of this enzyme. O.K. cloned multiple protein isoforms in *E. coli* and insect cells, developed protein purification and Se-Met labeling protocols, and contributed to crystallization and activity measurements. I.M. assisted with cloning of mutants, cell culture, protein purification, activity, and AUC assays. R.S. collected data

set from native protein crystal and helped with data collection strategies for selenium protein derivatives. K.R.M. and S.K wrote the paper with input from all authors.

## Additional information

**Competing interests:** The authors declare no competing financial interests.

