## [Peer Review File · Nature Communications]

Reviewers' comments:

Reviewer #1 (Remarks to the Author):

This is an interesting and potentially important study that describes the first crystal structure of the human iPLA2 β , which is one of two major splice variants of PLA2g6 gene. The results of this study revise the current expectations for the structure of this short splice variant, and demonstrate that instead of previously anticipated monomer, it can form a dimer through a close association of two individual catalytic domains. A single calmodulin is proposed to bind and allosterically inhibit both catalytic domains. The structural findings are novel, and may be important for better understanding the framework for catalytic activity and inhibitory CaM association with this specific variant of PLA2g6 gene. However, while the structure is intriguing, the current manuscript falls short of providing its rigorous structural analysis and structure-based interpretations of the protein function. It is also unfortunate that this study is presented in a rather superficial way: the manuscript offers very few details, figure legends lack substance, and multiple important claims made throughout this manuscript are not supported by rigorous structural or functional data. As a result, some solid conclusions appear to be heavily intermixed with speculations (see specific comments below), making it hard for the reader to separate evidence-based interpretations of the data from hypothetical assumptions.

Specific comments:

1. Completeness of crystallographic data at the specified resolution (4 Å) is only 34%, which is rather low and indicative of potentially poor quality of crystallographic data even within this (not high) resolution range. The entire model is based on these data, which makes it rather ambiguous, and may question the rigor of some bold statements and conclusions made in this manuscript. Further improvements of crystallographic data would be desirable.
2. The abstract (Lines 30-32) states that "These unique structural features identify molecular interactions which regulate cellular localization." However, from the manuscript it remains totally unclear which specific structural features (and how) can explain specific cellular localization(s) of the short iPLA2 β variant. There seem to be no data presented on the changes in cellular localization due to mutations of any specific residues to justify this statement.
3. A rather bold statement that "...ankyrin domains ideally positioned to target membrane proteins..." (lines 77-78) does not seem to be supported by any actual data: manuscript states (lines 97-98) that "the outer helix of AR1 is poorly ordered", and this helix (as well as the first 122aa of N terminus) have been omitted from the model. Without the detailed analysis of the structure of the N terminal part of iPLA2, and identification of the specific part(s) that can be responsible for interactions with membrane proteins, the statement in lines 77-78 seems to lack scientific rigor, and may be totally premature.
4. Also, there seem to be no strict experimental basis for the following statement (lines 224-228): "unexpected conformation of ANK domains found in the crystal structure suggests a protein-interaction function important for recognition of membrane proteins and for the specific localization of iPLA2 β on membrane surfaces. Elongated ANK domains can form highly specific docking surfaces for several different protein partners."
5. One of the potentially most problematic parts of the structure is the connector and interface between ANK and CAT domains, which authors suggest to be directly involved in transition between active and inactive states (Fig.5). It is shown with dashed line on cartoon in Fig. 1, and cannot be resolved in Fig.2. In Fig.2 the end of ANK9 domain (on the left) is simply hanging in air. This part of the structure is critical for mutual orientation of ANK and CAT domains, and need to be clearly resolved and presented in a separate figure, before any conclusions or speculations can be made.
6. It is unclear why "hydrophobic interface between the CAT and ANK domains is likely to permit conformational flexibility and rotation of the ANK domain around the interface" (lines 163-64). The reasons why "the relative orientation of the ANK and CAT domains ... is slightly different in the two monomers"... is also unclear, especially in view of the reported finding that "upon superposition of

CAT domains, the N-terminal sections of the ANK domains are shifted by $\sim 12 \text{ \AA}$ (Fig. S2E)".

7. Also, there seem to be no strict experimental basis for the following statement (lines 224-228): "unexpected conformation of ANK domains found in the crystal structure suggests a protein-interaction function important for recognition of membrane proteins and for the specific localization of iPLA2 β on membrane surfaces. Elongated ANK domains can form highly specific docking surfaces for several different protein partners."

8. Manuscript is poorly written, and it is even hard to understand whether the actual crystal structure obtained and described in this study represents CaM-bound ("closed", inactive), or the active state (a and b in Fig.5)? It seems to be crystallized with both iPLA2 and CaM present, but the structures are simply superimposed in the figures. So, does the identified structure represent inactive state (with CaM bound?)? Along the same lines of confusion about in which state (a or b in Fig.5) iPLA2 was actually crystallized and analyzed: it is stated that the crystals were soaked with Se-ATP (line 329), which suggests that crystal structure may have ATP bound to iPLA2 (like in Fig.5b), which corresponds to the active state rather than CaM-bound inactive state (Fig.5a). So, it is totally confusing which state (active or inactive, or both?) have been obtained in the reported crystals.

9. At that point, ANK transitions between active and inactive states seem to be a pure speculation. Without clear resolution of the linker between ANK and CAT domains, the suggested conformational changes in orientation of ANK domain (as in Fig.5) remain totally hypothetical and need to be clearly presented as such.

10. Although stated in the abstract (lines 31-32), there seem to be no direct data in this study that can elucidate how this new structure can explain the "role of neurodegenerative mutations and the function of iPLA2 β in the brain"

11. From the text of this manuscript it is almost impossible to understand which actual mutants were crystallized and analyzed. The list of tested mutants should be presented. Several mutations are mentioned throughout the manuscript, but without any specific information. Which (if any) mutants have been tested in this study that are associated with human PD, and which are related to INAD? It is unclear how mutants were selected, and what they have to do with pathophysiology of iPLA2 disease? The differences in catalytic activity of INAD and PD mutants are mentioned in the text, but the actual reasons for such differences are not addressed by this study.

Minor points:

- some figures are mis-referenced (line 178 Fig 3c, instead of 4c; line 185 Fig.3a instead of 4a;...),
- some mutants are mislabeled (W965E instead of presumably W695E ... in lines 184, 193, 436, ...)
- references need to accompany the statements like in lines 62-63
- Figure legends are cryptic and need to be extended to better describe the actual data.

Reviewer #2 (Remarks to the Author):

The manuscript of Malley et al presents the first crystal structure of calcium-independent phospholipase A2 (iPLA2 β or GVIA iPLA2). This enzyme is involved in multiple physiological and pathophysiological processes and thus is a target for the development of new medicinal agents.

They identified nine ankyrin repeats and they describe the formation of a dimer through CAT domains. In addition, they discuss the allosteric inhibition by calmodulin.

The results are very interesting and merit publication in Nature Communications. Thus, I recommend publication after revision.

Comments

1. The authors failed to cite the most important iPLA2 inhibitors (polyfluoroketones) described in literature (J. Med. Chem. 2010, 53, 3602-3610; Bioorg. Med. Chem. 2013, 21, 5823-5829), which are summarized in a very recent review (Exp. Opin. Ther. Patents 2017, 27, 217-225).

2. Using these small molecule inhibitors the importance of iPLA2 in experimental autoimmune encephalomyelitis (Brain 2009, 132, 1221-1235) and diabetes type 1 (Diabetes 2015, 64, 541-

554) as well as the importance of developing inhibitors as therapeutic agents is demonstrated. The authors have to include the appropriate references.

3. As the authors state at the end of the Discussion, "The structure should also facilitate the design of small molecule modulators of iPLA2 β for therapeutic purposes." Up to now, the model based on molecular modeling has been successfully used in designing small molecule inhibitors (J. Med Chem. 2016, 59, 4403-4414). The authors have to indicate the differences (if any) of the catalytic domain between the structure described in this manuscript and the previous model and explain how the present structure would facilitate the design of small molecule inhibitors.

Reviewer #3 (Remarks to the Author):

This manuscript describes the moderate-resolution structure of iPLA2 β that reveals a dimerization interface via the catalytic domain. The dimer is supported with biochemical data where disruption of the dimerization interface with W695E reveals changes in analytical ultracentrifugation (AUC). Their hypothesis is thus that CaM locks iPLA2 β in the crystallized conformation where the active sites would be prevented from binding to the membrane. There is biochemical data to support this hypothesis, specifically the weaker FAM-CaM association with W695E. Overall, it is the dimerization and the orientation of the structure that represents a new advance in the field.

The manuscript is hard to read and tends to list fact after fact instead of reading more like a story with transitions. It is often unclear at first glance what the authors did and how they did it versus what was previously published. As examples, line 193 mentions a mutation that is not described anywhere else except the methods, and there are mentions of both W965E and W695E, but assumed to be W695E.

One major concern is that CaM appears to be in the crystallization condition yet not observed in the crystal structure, and there is no discussion about this. Is the unit cell big enough and the packing arrangement such a way in order to accommodate CaM? Have the authors tried to run the crystals on a gel to see if CaM is indeed in the crystallized structure? Or can they form the same crystals without CaM? If indeed CaM is critical in stabilizing this conformation, one would expect there to be electron density for it. Furthermore, their model with CaM as shown in Figure 4d and 4e is not very convincing. It looks like the authors just put CaM on top of their structure. If they want to show a model like this, some sort of docking would make more sense. In the structure they reference, the 1-9-14 motif is more inside the CaM, which is not how they show it here. Also, CaM exists in a variety of conformations so it would be very difficult to propose a structure without it being experimentally determined. The model in 5a is probably sufficient for their purposes. Instead of the structural model in Fig 4, perhaps they could just show or describe the measurements they did between the two proposed CaM binding sites, etc. Lastly, the description of CaM binding motifs and peptide binding is confusing. The IQ motif is a more common CaM binding motif that the authors seem to dismiss.

The authors mention previous data where iPLA2 β was determined to have higher order oligomers, however, they don't give enough data/description to convince readers that iPLA2 β is a dimer in solution. This reviewer would like to see a better description of predicted vs experimentally determined molecular weights for the AUC experiments and how the elongated conformation would hinder the analysis.

The manuscript is confusing about if iPLA2 β is able to bind membranes at all when in complex with CaM? It is unclear if the membrane in Fig 2B is just to show the membrane orientation or if the N-terminus is hypothesized to bind membranes. If so, this should be made clear. If the hypothesis is indeed that no membrane binding should occur with CaM inhibition, membrane binding experiments such as liposome cosedimentation would help strengthen this hypothesis.

For Figure 4C, since this data is critical to their hypothesis, it would be better to show error bars to know if it is significant or not (even if the data are normalized to bring individual experiments in the same range).

Other minor comments:

In intro – Describe what a canonical ANK domain looks like (two helices with linker loops, how oriented?)

Line 58 – How is patatin homologous? Percent identity?

Line 82 – Mention that it's a splice variant and mention here that its missing 54 amino acid proline rich region.

Line 82 – In description of structure – mention which residue are visible see in the structure (and CaM), as well as RMSD for structures used for molecular replacement and their overlays together (one mentioned later in Fig S2B but at minimum should be referenced here).

Line 99 – change to just SH-iPLA2 β

Line 141 – show how W695 forms stacking interaction in a figure and reference here.

Line 206 – missing the "i" in ipla2beta

Last paragraph – missed opportunity to map disease variants on the structure and speculate how they are disease-causing. This would certainly elevate the manuscript.

Fig1B – use dashed lines to link part that isn't modeled

Fig 1 – Ray trace these images at higher resolution. Depth cue would help too.

We are grateful to all reviewers for appreciating the importance and novelty of our data and for the detailed and critical evaluation of the manuscript. We significantly changed our presentation, removing speculations from the Results section and making a more concise Discussion section. We have added descriptions of disease mutations, a more detailed comparison with the patatin structure, extended figure legends and added missing references.

Specific answers to all questions are below.

Reviewers' comments:

Reviewer #1 (Remarks to the Author):

This is an interesting and potentially important study that describes the first crystal structure of the human iPLA2 β , which is one of two major splice variants of PLA2g6 gene. The results of this study revise the current expectations for the structure of this short splice variant, and demonstrate that instead of previously anticipated monomer, it can form a dimer through a close association of two individual catalytic domains. A single calmodulin is proposed to bind and allosterically inhibit both catalytic domains. The structural findings are novel, and may be important for better understanding the framework for catalytic activity and inhibitory CaM association with this specific variant of PLA2g6 gene. However, while the structure is intriguing, the current manuscript falls short of providing its rigorous structural analysis and structure-based interpretations of the protein function. It is also unfortunate that this study is presented in a rather superficial way: the manuscript offers very few details, figure legends lack substance, and multiple important claims made throughout this manuscript are not supported by rigorous structural or functional data. As a result, some solid conclusions appear to be heavily intermixed with speculations (see specific comments below), making it hard for the reader to separate evidence-based interpretations of the data from hypothetical assumptions.

We significantly improved the presentation, added more details and additional description to figure legends and figures. The major result of the work is the structure itself and its interpretation, including a rational hypothesis based on comparison with similar structural motifs of known functions. In the case of the first solved structure of a protein, particularly when revealing novel and unexpected structural features, it is usually beyond the scope of a single manuscript to verify every structure-based hypothesis in vitro, in cells, or in animal model.

Specific comments:

1. Completeness of crystallographic data at the specified resolution (4 Å) is only 34%, which is rather low and indicative of potentially poor quality of crystallographic data even within this (not high) resolution range. The entire model is based on these data, which makes it rather ambiguous, and may question the rigor of some bold statements and conclusions made in this manuscript. Further improvements of crystallographic data would be desirable.

A poor completeness of data with intensities above 1 sigma refers only to a highest resolution data shell and not to the entire data set. We added a better explanation in the Methods.

Electron density maps are of a high quality and are sufficient to reliably build an atomic resolution structure. We deliberately avoided modeling of structural parts with poorly defined electron density. All the structural elements were carefully verified with the rigorous quality control methods developed over the past 20 years. A detailed report on the quality of the structure was submitted with the paper.

The resolution of the structure is well within acceptable range. There are more than 600 crystal structures with resolution worse than 4.0 Å and more than 200 crystal structures resolved at resolution worse than 4.5 Å, most of which are published in top peer-reviewed journals.

Furthermore, the main result from this work is a discovery of unusual, novel arrangement of the CAT and ANK domains. The claim of such configuration can be made with lower resolution data than ours.

2. The abstract (Lines 30-32) states that “These unique structural features identify molecular interactions which regulate cellular localization.” However, from the manuscript it remains totally unclear which specific structural features (and how) can explain specific cellular localization(s) of the short iPLA₂β variant. There seem to be no data presented on the changes in cellular localization due to mutations of any specific residues to justify this statement.

This statement refers to the unique (and the only known) function of ankyrin repeats, often described as antibody-like molecules due to highly specific protein recognition properties. We have added additional explanation and description of AR to the text. We also stated that extended proline-rich loops, some of which are present in both long and short variants, are often found to mediate protein recognition (new reference 52).

3. A rather bold statement that “...ankyrin domains ideally positioned to target membrane proteins...” (lines 77-78) does not seem to be supported by any actual data: manuscript states (lines 97-98) that “the outer helix of AR1 is poorly ordered”, and this helix (as well as the first 122aa of N terminus) have been omitted from the model. Without the detailed analysis of the structure of the N terminal part of iPLA₂, and identification of the specific part(s) that can be responsible for interactions with membrane proteins, the statement in lines 77-78 seems to lack scientific rigor, and may be totally premature.

Determining structures of the first AR and of the N-terminal domain will certainly be beneficial for studies of iPLA₂β and may suggest a more specific mechanism of potential interactions. However, this will not alter the orientation, observed for the first time in our work. We are not claiming the detailed inter-atomic interactions between the ANK domains and a membrane. Rather, we are presenting a newly observed orientation of ANK domains which are “positioned” for such interaction. We feel that our crystallographic data are sufficient to make this specific claim.

Specific protein recognition, including interaction with membrane proteins, as in case of canonical ankyrins Ankr, AnkB and AnkG, is the only known function of ARs found in thousands of different proteins. In the absence of any other proposed mechanism of iPLA₂β tissue-specific membrane localization, this seems to be the only plausible structure-based hypothesis so far. Again, this does not preclude involvement of other structural parts with the membrane or membrane proteins.

We have added additional references (49,50,74-76) to the related publications.

4. Also, there seem to be no strict experimental basis for the following statement (lines 224-228): “unexpected conformation of ANK domains found in the crystal structure suggests a protein-interaction function important for recognition of membrane proteins and for the specific localization of iPLA₂β on membrane surfaces. Elongated ANK domains can form highly specific docking surfaces for several different protein partners.”

This is correct. This structure only suggests but does not prove such a hypothesis. However, in the absence of other experimental support, we believe it is important to use novel structural data to present a rational hypothesis addressing significant gap in the field. As we stated in the introduction, the protein interaction properties of iPLA₂β have not been extensively studied thus far. This is a significant gap in case of a protein, more than half of which is comprised of canonical protein recognition domains, which was shown in many publications to have a tissue-specific membrane localization, and which does not contain known membrane binding domains (such as the C2 domain of cPLA₂, for example). We believe that structure-based hypotheses will stimulate and direct more extensive investigation of such potential interactions, critical for understanding the mechanism and function of this enzyme.

We have added references to the text describing structures where relatively short ANK domains specifically interact with several different molecules (75,76).

5. One of the potentially most problematic parts of the structure is the connector and interface between ANK and CAT domains, which authors suggest to be directly involved in transition between active and inactive states (Fig.5). It is shown with dashed line on cartoon in Fig. 1, and cannot be resolved in Fig.2. In Fig.2 the end of ANK9 domain (on the left) is simply hanging in air. This part of the structure is critical for mutual orientation of ANK and CAT domains, and need to be clearly resolved and presented in a separate figure, before any conclusions or speculations can be made.

We have provided an additional figure zooming in on the region of the ANK-CAT linker. Indeed, this conformation was extremely puzzling for us and we did analyze all potential connections between ANK and CAT in the crystal. There are no other possibilities to connect the two domains in this crystal lattice. Besides, every ANK domain in the crystal forms an identical interface with the CAT domain with its N-terminus pointing towards the membrane-binding surface of the CAT domain.

We would like to reiterate that our main claim is the unusual configuration of the CAT and ANK domains and our structural data allow such claim, even without resolving the atomic details of how such configuration is achieved.

6. It is unclear why “hydrophobic interface between the CAT and ANK domains is likely to permit conformational flexibility and rotation of the ANK domain around the interface” (lines 163-64).

This statement refers to the physical nature of hydrophobic interactions which are not specific. This non-specificity allows structural mobility of protein domains connected through hydrophobic interfaces, which is well demonstrated in multiple structural and biophysical studies, i.e. rotation of F1-ATPase around the stalk (PMID: 8065448, 9069291) or rotation of domains in resolvase (PMID: 16807292).

The reasons why “the relative orientation of the ANK and CAT domains ... is slightly different in the two monomers” ... is also unclear, especially in view of the reported finding that “upon superposition of CAT domains, the N-terminal sections of the ANK domains are shifted by ~12 Å (Fig. S2E)”.

This is an experimental result. It refers to an actual difference in conformation of ANK domains relatively to CAT domains between A and B monomers observed in the crystal structure. Two molecules of the dimer were overlaid using only coordinates of the CAT domains. Changes in the resulting orientation of the ANK domains were measured.

7. Also, there seem to be no strict experimental basis for the following statement (lines 224-228): “unexpected conformation of ANK domains found in the crystal structure suggests a protein-interaction function important for recognition of membrane proteins and for the specific localization of iPLA₂β on membrane surfaces. Elongated ANK domains can form highly specific docking surfaces for several different protein partners.”

See above responses to questions 2-4.

8. Manuscript is poorly written, and it is even hard to understand whether the actual crystal structure obtained and described in this study represents CaM-bound (“closed”, inactive), or the active state (a and b in Fig.5)? It seems to be crystallized with both iPLA₂ and CaM present, but the structures are simply superimposed in the figures. So, does the identified structure represent inactive state (with CaM bound?)?

We would like to thank the reviewer for pointing out this confusing part. We have clarified this statement in the revised manuscript. The structure is of free iPLA₂β without CaM.

We clarified this in the text and added to the Methods that the initial crystals were obtained in the absence of CaM. During later attempts to co-crystallize the enzyme with CaM, identical crystals were obtained, and the SDS PAGE analysis of crystals indicated the absence of CaM. Since a higher concentration of iPLA₂β could be achieved in the presence of CaM, and CaM did not interfere with crystallization itself, we continued protein preparation for crystallization in presence of CaM. This approach saved us large quantities of the protein, produced from insect cells during crystallization trials.

The crystal packing is not compatible with the proposed CaM-bound complex, and we believe that the crystal structure represents an active conformation. In fact, the active site is wide open and can accommodate long unsaturated phospholipids.

Along the same lines of confusion about in which state (a or b in Fig.5) iPLA₂ was actually crystallized and analyzed: it is stated that the crystals were soaked with Se-ATP (line 329), which suggests that crystal structure may have ATP bound to iPLA₂ (like in Fig.5b), which corresponds to the active state rather than CaM-bound inactive state (Fig.5a). So, it is totally confusing which state (active or inactive, or both?) have been obtained in the reported crystals.

There is only one conformation state crystallized. Soaking with SeATP did not alter the overall conformation of domains.

9. At that point, ANK transitions between active and inactive states seem to be a pure speculation. Without clear resolution of the linker between ANK and CAT domains, the suggested conformational changes in orientation of ANK domain (as in Fig.5) remain totally hypothetical and need to be clearly presented as such.

Yes, this is a hypothesis. Based on the unstructured conformation of the relatively long linker and the hydrophobic interaction between ANK and CAT domains, it is highly possible that ANK can adopt alternative conformations. Further structural studies are required to investigate this issue.

10. Although stated in the abstract (lines 31-32), there seem to be no direct data in this study that can elucidate how this new structure can explain the “role of neurodegenerative mutations and the function of iPLA₂ β in the brain”

We have added a new figure 6 illustrating locations and a new paragraph describing the potential role of previously discovered PARK and INAD mutations in the structure.

11. From the text of this manuscript it is almost impossible to understand which actual mutants were crystalized and analyzed. The list of tested mutants should be presented. Several mutations are mentioned throughout the manuscript, but without any specific information. Which (if any) mutants have been tested in this study that are associated with human PD, and which are related to INAD? It is unclear how mutants were selected, and what they have to do with pathophysiology of iPLA₂ disease?

Only WT CHO SH-iPLA₂ β with a C-terminal His-tag and the I701D mutant (in case of SeMet crystals) were crystallized so far. Both have identical conformations and final refinement was performed using data from the WT protein.

Negative crystallization experiments are inconclusive and information about variants that were not crystallized is not usually reported.

The differences in catalytic activity of INAD and PD mutants are mentioned in the text, but the actual reasons for such differences are not addressed by this study.

We have added descriptions of INAD and PD mutants, for which biochemical activity was previously tested. As we stated in the text, more quantitative measurements of the enzymatic activity and of macromolecular interactions are required to fully understand the effect of each mutation on the structure and function.

Minor points:

- some figures are mis-referenced (lane 178 Fig 3c, instead of 4c; line 185 Fig.3a instead of 4a;...),

Fixed

- some mutants are mislabeled (W965E instead of presumably W695E ... in lines 184, 193, 436, ...)

Fixed

- references need to accompany the statements like in lines 62-63

We have added references 49, 50, 52 and, later, 74-76.

- Figure legends are cryptic and need to be extended to better describe the actual data.

Thank you for pointing out all these errors. We fixed all the errors, added references and expanded the figure legends.

Reviewer #2 (Remarks to the Author):

The manuscript of Malley et al presents the first crystal structure of calcium-independent phospholipase A2 (iPLA2 β or GVIA iPLA2). This enzyme is involved in multiple physiological and pathophysiological processes and thus is a target for the development of new medicinal agents. They identified nine ankyrin repeats and they describe the formation of a dimer through CAT domains. In addition, they discuss the allosteric inhibition by calmodulin.

The results are very interesting and merit publication in Nature Communications. Thus, I recommend publication after revision.

Comments

1. The authors failed to cite the most important iPLA2 inhibitors (polyfluoroketones) described in literature (J. Med. Chem. 2010, 53, 3602-3610; Bioorg. Med. Chem. 2013, 21, 5823-5829), which are summarized in a very recent review (Exp. Opin Ther. Patents 2017, 27, 217-225).

We have added references 39-42.

2. Using these small molecule inhibitors the importance of iPLA2 in experimental autoimmune encephalomyelitis (Brain 2009, 132, 1221-1235) and diabetes type 1 (Diabetes 2015, 64, 541-554) as well as the importance of developing inhibitors as therapeutic agents is demonstrated. The authors have to include the appropriate references.

We have added these references and additional description of the current progress in introduction.

3. As the authors state at the end of the Discussion, "The structure should also facilitate the design of small molecule modulators of iPLA2 β for therapeutic purposes." Up to now, the model based on molecular modeling has been successfully used in designing small molecule inhibitors (J. Med Chem. 2016, 59, 4403-4414). The authors have to indicate the differences (if any) of the catalytic domain between the structure described in this manuscript and the previous model and explain how the present structure would facilitate the design of small molecule inhibitors.

We added 1) a paragraph describing differences between the active sites of iPLA₂ β and patatin and figures S3 b,c, important for design of the mechanism-based inhibitors, and 2) a description of auxiliary domains and motifs that can be attractive for more specific inhibition of either enzymatic activity or regulatory interactions.

Reviewer #3 (Remarks to the Author):

This manuscript describes the moderate-resolution structure of iPLA2 β that reveals a dimerization interface via the catalytic domain. The dimer is supported with biochemical data where disruption of the dimerization interface with W695E reveals changes in analytical ultracentrifugation (AUC). Their hypothesis is thus that CaM locks iPLA2 β in the crystallized conformation where the active sites would be prevented from binding to the membrane. There is biochemical data to support this hypothesis,

specifically the weaker FAM-CaM association with W695E. Overall, it is the dimerization and the orientation of the structure that represents a new advance in the field.

The crystal structure is of CaM-free protein and was obtained both in the presence and absence of CaM in the crystallization mix. We rewrote the methods to remove the confusion (see also response to reviewer #1, question 8).

The manuscript is hard to read and tends to list fact after fact instead of reading more like a story with transitions. It is often unclear at first glance what the authors did and how they did it versus what was previously published.

We significantly rearranged the presentation to make the story clearer.

As examples, line 193 mentions a mutation that is not described anywhere else except the methods, and there are mentions of both W965E and W695E, but assumed to be W695E.

We have corrected this error and added comments and a figure S5c explaining the rationale behind this mutation.

One major concern is that CaM appears to be in the crystallization condition yet not observed in the crystal structure, and there is no discussion about this. Is the unit cell big enough and the packing arrangement such a way in order to accommodate CaM? Have the authors tried to run the crystals on a gel to see if CaM is indeed in the crystallized structure? Or can they form the same crystals without CaM? If indeed CaM is critical in stabilizing this conformation, one would expect there to be electron density for it.

CaM is not in the crystal (see comments above) and it was confirmed with SDS page analysis. With respect to catalytic domains, the structure seems to be in an open conformation and the active sites are accessible for substrate binding (new Figs. S3 b,c). We included this discussion in the text. It is unclear whether and/or how much the conformation of ANK domains, as seen in crystal, will interfere with interaction of the CAT domain with membrane.

Furthermore, their model with CaM as shown in Figure 4d and 4e is not very convincing. It looks like the authors just put CaM on top of their structure. If they want to show a model like this, some sort of docking would make more sense. In the structure they reference, the 1-9-14 motif is more inside the CaM, which is not how they show it here. Also, CaM exists in a variety of conformations so it would be very difficult to propose a structure without it being experimentally determined. The model in 5a is probably sufficient for their purposes. Instead of the structural model in Fig 4, perhaps they could just show or describe the measurements they did between the two proposed CaM binding sites, etc.

The purpose of the figure is indeed only to illustrate the possibility of such interactions, judging purely by the distance between peptide-binding sites of two domains in CaM and 1-9-14 motifs. We did not perform real docking with CaM because a large part of 1-9-14 motif does not have a well-defined electron density, likely due to flexibility of this loop in the absence of CaM. Judging from an open conformation of the active sites, the crystal structure represents an active conformation and CaM binding may require additional conformational changes of the dimer. Consequently, docking will be too speculative at this point.

Lastly, the description of CaM binding motifs and peptide binding is confusing. The IQ motif is a more common CaM binding motif that the authors seem to dismiss.

We agree that IQ is a common motif, however, in the structure this motif is buried and forms one of the central β -strands of the core domain. A surface-exposed helical conformation of this region would require complete unfolding of the CAT domain.

The authors mention previous data where iPLA₂ β was determined to have higher order oligomers, however, they don't give enough data/description to convince readers that iPLA₂ β is a dimer in solution. This reviewer would like to see a better description of predicted vs experimentally determined molecular weights for the AUC experiments and how the elongated conformation would hinder the analysis.

We added expected and experimental values.

The manuscript is confusing about if iPLA₂ β is able to bind membranes at all when in complex with CaM?

It is unclear if the membrane in Fig 2B is just to show the membrane orientation or if the N-terminus is hypothesized to bind membranes. If so, this should be made clear.

Yes, the membrane is just to orient the viewer. We added clarification to the figure legend.

If the hypothesis is indeed that no membrane binding should occur with CaM inhibition, membrane binding experiments such as liposome cosedimentation would help strengthen this hypothesis.

This is an excellent suggestion to verify our weak hypothesis. We did perform co-sedimentation experiments with liposomes identical to those used for activity measurements, as well as sedimentation velocity AUC with DMPC nano-disks using either WT or the S495A inactive mutant. No strong stable association of the enzyme with liposomes or the bilayer was observed in either experiment under conditions identical to those in the activity assays without CaM. The fact that CaM significantly affects protein solubility further complicates interpretation of data in such experiments.

The affinity of iPLA₂ β to bilayer is an important mechanistic factor. However, the estimation of weak transient interaction, as it seems to be the case, will require more sensitive quantitative methods. The complexity and the significance of the topic warrant multiple experiments, optimization of various methods and comparative analysis of different variants and complexes, which will exceed the scope of the current manuscript. Since this was not a major concern of the reviewer, we decided to omit preliminary data.

Meanwhile, we removed this speculation from the text, first, since the long variant is purified in the membrane-bound state and is inactive in the presence of CaM/Ca²⁺, and, second, since this hypothesis was based only on the fact that the linker and the putative CaM-binding sites are simply at the same side of the protein surface. In fact, one is close to the dimerization interface, while the other faces in outwards direction.

For Figure 4C, since this data is critical to their hypothesis, it would be better to show error bars to know if it is significant or not (even if the data are normalized to bring individual experiments in the same range).

We have fixed the data presentation. In fact, the error bars were present in original figure, but were hidden behind marks.

Other minor comments:

In intro – Describe what a canonical ANK domain looks like (two helices with linker loops, how oriented?)

Added

Line 58 – How is patatin homologous? Percent identity?

Added

Line 82 – Mention that it's a splice variant and mention here that its missing 54 amino acid proline rich region.

Fixed

Line 82 – In description of structure – mention which residue are visible see in the structure (and CaM), as well as RMSD for structures used for molecular replacement and their overlays together (one mentioned later in Fig S2B but at minimum should be referenced here).

Line 99 – change to just SH-iPLA2 β

Fixed

Line 141 – show how W695 forms stacking interaction in a figure and reference here.

New figure S5c

Line 206 – missing the “i” in ipla2beta

Fixed

Last paragraph – missed opportunity to map disease variants on the structure and speculate how they are disease-causing. This would certainly elevate the manuscript.

We have added the description of mutants

Fig1B – use dashed lines to link part that isn't modeled

Fig 1 – Ray trace these images at higher resolution. Depth cue would help too.

Fixed.

Thank you again for a detailed critique. We fixed the errors mentioned in all minor comments.

Reviewers' comments:

Reviewer #1 (Remarks to the Author):

This reviewer understands the author's opinion and appreciates the efforts in revising this manuscript, which now provided more clarifications and details. However, the revision failed to improve the scope and quality of the studies. Responses to specific questions were mostly argumentative, did not provide clear experimental evidence to support the author's claims, and did not address concerns about largely overreaching conclusions that did not seem to be substantiated by actual data in this manuscript. References to 600+ papers with limited resolution structures published in the past, or to the published studies of other proteins that may "resemble" parts of iPLA2g6 did not strengthen the current study. Instead, they confirmed the limitations of the new experimental data presented in this study that (using authors own words in rebuttal, point 4) "only suggests but does not prove hypothesis" for the structure-function (or pathogenic dysfunction) of this complex protein.

New major concerns arise after the original data were clarified, especially in view of intrinsic contradictions, questionable assumptions/conclusions and largely overreaching claims.

Specifically:

(1) Functionality and physiological relevance of the experimentally obtained crystallographic structure: Revised manuscript (and rebuttal) now clearly states that crystal structure (presented in Fig.2) represents an active state of iPLA2. Indeed, it seems reasonable, as it is CaM-free and has active CAT site in open conformation, which is schematically illustrated in Fig.5b. However, ANKs orientation tells a different story, as it shows sharp angle towards membrane, which (as described in Fig.5a) is proposed to represent inactive state that precludes active site access to lipid bilayer. So, the actual crystal structure (in Fig.2) does not seem to fully represent either active (Fig.5b) or inactive (5a) state. If one will assume that the actual crystal structure (in Fig.2) represents active state (which is logical in view of no CaM binding and active site being fully open), then CAT domain will unlikely to be able to reach membrane to perform its physiological function because of the prohibitory orientation of ANKs. Thus, the structure of the actually crystallized protein (in Fig.2) does not support (and may directly contradict) the main hypothesis of this manuscript about functional transitions, as illustrated in Fig.5. It looks like the short iPLA2 variant may have a much better chance to reach the membrane and perform its catalytic activity... if it will be in monomeric state, which raises a major concern regarding monomeric versus dimeric state under physiological conditions. A more careful look at the figures S4a and Fig.4a raised additional questions about whether dimerization of short iPLA2 (captured in the crystal) is physiological, or could be triggered by its concentration under experimental conditions. Indeed, crosslinking experiment in Fig.S4a, and sedimentation velocity distribution in Fig.4a clearly show that short iPLA2 can form monomers and dimers in solution, but concentration-dependency (or rather the lack of it) in Fig.S4a raises more concerns. In such experiment one will expect that increase in iPLA2 concentration will increase the probability of forming dimers. However, while more protein can be clearly detected at 0 and 0.1mM BS3 in 150nM iPLA2g6 samples, almost the same ratio between monomers and dimers can be seen at 1-10mM BS3 with both 50 and 150nM iPLA2 (it even looks like there may be more % of dimers in 50nM). It is unclear if tetramers could be formed (the current blot is limited to 220kDa). To resolve these concerns, full range of MWs (covering potential tetramers) need to be presented in crosslinking experiments, and careful quantitative analysis of the ratio between monomers and dimers at different protein concentrations should be performed. It will be also very important to demonstrate that short iPLA2 indeed comes as a dimer under physiological conditions in native gels. In the absence of such detailed analysis, the question remains, whether dimers seen in crystals are present and relevant for cellular function under physiological conditions. It is also worrisome that IQ domain was found to be buried within the CAT domain and inaccessible for CaM binding, which contradicts earlier studies that convincingly demonstrated the role of IQ in CaM binding and iPLA2 inhibition (some of which were published by several authors of the current

study). So, there seem to be an internal contradiction, and either the earlier studies/results were incorrect (and may need retraction), or one can expect that the iPLA2 structure should accommodate accessibility of IQ for CaM binding/inhibition.

(2) Another major concern is the continuous confusion between short and long splice variants and the lack of their clear distinction in this manuscript in terms of their potentially different structure, function, cellular localization and pathophysiology. As an example, comparison of the structure of the short variant in current manuscript to the findings reported in {62} (Larsson et al, 1998), {81} (Engel et al, 2010), or {24} (Zhou et al, 2016) sounds inappropriate, because all these (and some other) papers describe the properties of the full length 806-amino acid protein with a calculated molecular mass of 88 kDa, and not its short splice variant (752aa). The authors failed to consider the fact that addition of 54-amino acid proline-rich insertion in the linker between ANK and CAT domains (residues 395–449) can significantly change their mutual orientation and dramatically affect the overall structural properties of the molecule, its cellular location and physiological function. Thus, all the claims and conclusions of the current study should be very clearly limited to the short splice variant. The lack of discrimination between specific variants can further fuel ongoing confusion regarding the cellular localization, physiological function and pathogenic dysfunction that can be different for different splice variants of iPLA2.

(3) Significant weakness of this manuscript is abundant references to iPLA2 role in different pathological conditions with apparent lack of explanation of the structure/functional roots for the differences between iPLA2 mutants association with distinct neurodegenerations. While a new figure 6 is appreciated, only few hand-picked mutations are shown, and revised manuscript does not provide sufficient explanation for the structural differences that could lead to association with INAD, PD or other conditions. For example, the current model does not seem to explain why R741 mutation to W leads to the loss of catalytic activity and INAD, while its mutation to Q (R741Q) does not affect catalytic activity, and yet leads to PD. It is also unclear how other mutations can affect catalytic activity or other properties of this protein. Please, note that there is a mistake in the well-known PD-associated mutation in position 747: it is R747W, and not R747Q! Surprisingly, this mutation is not even mentioned in discussion, and it is totally unclear which aspect of iPLA2 function such mutation can affect, and how it can be linked to PD.

A minor comment: labeling of genetic mutations using sequence of the shorter splice variant is a very unusual and confusing. To avoid confusion, disease-associated mutations are expected to represent the actual mutations within the gene sequence, and aa substitutions should be labeled based on the full length protein, not by aa numbers in the splice variants.

(4) The claims for the ANK interactions with membrane proteins remain totally hypothetical (based on analogies to other proteins and generalized predictions, not on actual data), so, all such claims and speculations should be removed from Results and limited to Discussion.

Reviewer #3 (Remarks to the Author):

Thank you for the revised manuscript. It is much better written and more clear, and most issues have been addressed satisfactorily. I would recommend publication after the following changes are made.

Line 29 in the abstract leads the reader to believe that the calmodulin was observed in the crystal structure or that there are conclusive data. It should say something less definitive, such as "the structure combined with biochemical evidence suggest that a single calmodulin is able to bind..", or "we hypothesize that a single calmodulin is able to bind.."

Minor comments/edits:

Line 61 – edit this phrase "The AR is 33-amino acid long"

Line 91 – It would help to specify that the short variant is lacking the proline rich region and reference Fig. 1A here

Line 92 – write out SAD

Line 93 – Thank you for adding the homology with patatin, but that sentence reads a bit funny – the CAT domain is 32% homologous with patatin? The “four ARs of an ankryin-R protein” is something else but no homology is referenced? I would append this to the end of the previous sentence and change the period to a colon “: patatin, which is 32% homologous with the CAT domain, and four ARs of an ankryin-R protein”. However, I would also change homologous, which is vague, to similar or identical depending on which it is.

Line 101 – write out RMSD, change to read “an RMSD of 3.1 Å” (RMSD unit is not squared)

Line 281 – change to with the CAT domain

Line 387 – could you add the PDB codes for the proteins you used for molecular replacement along with the references, especially since four ARs of an ankryin-R proteins is vague.

Fig. 3 – uses both polyG and GGGVKG labeling, I would just pick one label

We are grateful to reviewers for a detailed evaluation of the revised manuscript, for appreciating our results and for additional suggestions and for highlighting remaining errors. However, several comments of the reviewer #1 seems to be based on misinterpretation of actual statements or on taking specific supplementary data outside of the context of main results. Below are detailed answers to each critique point.

Importantly, we included an additional reference describing iPLA2b interaction with multiple membrane proteins, including calnexin, which provides additional strong support for our hypothesis about interaction of iPLA2b with membrane proteins. Correspondingly, there are several additions to the introduction and discussion. The model in Figure 5 is updated to illustrate previously published iPLA2b interactions.

Reviewer #1 (Remarks to the Author):

This reviewer understands the author's opinion and appreciates the efforts in revising this manuscript, which now provided more clarifications and details. However, the revision failed to improve the scope and quality of the studies. Responses to specific questions were mostly argumentative, did not provide clear experimental evidence to support the author's claims, and did not address concerns about largely overreaching conclusions that did not seem to be substantiated by actual data in this manuscript.

The manuscript describes the first crystal structure of the full-length protein (short variant, 752 aa) and biochemical and mutagenesis data confirming oligomerization and calmodulin interaction mechanisms important for the protein function and regulation. These data, without a further expanding the scope of the work, represent a major advance in the field and it is difficult to overstate the importance of timely dissemination of the manuscript and coordinates of the structure. This goes along the accepted practice for publishing first crystal structures of difficult multi-domain proteins without answering definitively all questions on function and mechanism.

Similarly to novel sequencing data, 3D structure is an extremely informative tool to predict mechanistic features based on comparison with well-studied proteins. Our hypotheses are based on previously published functional and biochemical studies of iPLA2b and the comparison with a significant body of previously published homologous structures with known mechanism of function. The only discrepancy between our model and previous models is its dimeric versus a tetrameric form, which we carefully explained (see additional comments below).

Functional evaluation of these hypotheses in multiple iPLA2b-dependent signaling pathways is beyond the scope of a single publication.

References to 600+ papers with limited resolution structures published in the past, or to the published studies of other proteins that may "resemble" parts of iPLA2g6 did not strengthen the current study.

Reference to 600+ publications of lower resolutions structures was not meant to strengthen the current study. Rather, it was meant to highlight its validity by drawing an attention to the fact that publishing a moderate to lower resolution structures of difficult-to-crystallize multi-domain proteins is

an accepted practice and these structures are intended to foster further research and not necessarily to answer all functional questions.

Instead, they confirmed the limitations of the new experimental data presented in this study that (using authors own words in rebuttal, point 4) “only suggests but does not prove hypothesis” for the structure-function (or pathogenic dysfunction) of this complex protein.

The authors strongly disagree with the reviewer’s rejection of a power of comparative structural and sequence analysis. Comparative analysis is an extremely powerful tool in structural biology, invaluable for generating testable hypotheses to facilitate mechanistic and functional studies of proteins.

New major concerns arise after the original data were clarified, especially in view of intrinsic contradictions, questionable assumptions/conclusions and largely overreaching claims.

We included an additional reference describing interaction of iPLA2b with more than a dozen membrane proteins including detailed evaluation of iPLA2b interaction with the transmembrane ER protein calnexin. Previous studies, cited in the initial submission, document the interaction of the ANK domain with other cellular proteins. Combined with hundreds of well-documented protein-protein interactions of ARs, these data strongly support our hypotheses.

There are no intrinsic contradictions in the experimental results and hypotheses.

Specifically:

(1) Functionality and physiological relevance of the experimentally obtained crystallographic structure: Revised manuscript (and rebuttal) now clearly states that crystal structure (presented in Fig.2) represents an active state of iPLA2. Indeed, it seems reasonable, as it is CaM-free and has active CAT site in open conformation, which is schematically illustrated in Fig.5b. However, ANKs orientation tells a different story, as it shows sharp angle towards membrane, which (as described in Fig.5a) is proposed to represent inactive state that precludes active site access to lipid bilayer. So, the actual crystal structure (in Fig.2) does not seem to fully represent either active (Fig.5b) or inactive (5a) state. If one will assume that the actual crystal structure (in Fig.2) represents active state (which is logical in view of no CaM binding and active site being fully open), then CAT domain will unlikely to be able to reach membrane to perform its physiological function because of the prohibitory orientation of ANKs.

This critique distorts the actual description of our structural results and discussion. We only claim that the CAT domain is in open active conformation, not the entire protein (lines 106,107,215-217. The numbering corresponds to the previous resubmission). We specifically emphasize in the text that the ANK domain can adapt an alternative conformation upon interaction with the membrane (lines 275-284), we discussed the mechanism of such conformational changes and we even illustrated it by showing differences in orientation of ANK domains in two monomers of the dimer in the current crystal structure.

Thus, the structure of the actually crystallized protein (in Fig.2) does not support (and may directly contradict) the main hypothesis of this manuscript about functional transitions, as illustrated in Fig.5.

The hypothetical model in Figure 5 is not a low resolution drawing of the crystal structure. The figure illustrates two hypothetical functional conformational states of the protein, suggested by the structural data combined with the previously published data. It also illustrates published macromolecular interactions of iPLA2b. We added an additional explanation that the protein can adopt multiple conformations, the functional significance of which can be investigated in future studies. Our structure-based hypothesis provides a rational foundation for future functional studies.

It looks like the short iPLA2 variant may have a much better chance to reach the membrane and perform its catalytic activity... if it will be in monomeric state, which raises a major concern regarding monomeric versus dimeric state under physiological conditions. A more careful look at the figures S4a and Fig.4a raised additional questions about whether dimerization of short iPLA2 (captured in the crystal) is physiological, or could be triggered by its concentration under experimental conditions. Indeed, crosslinking experiment in Fig.S4a, and sedimentation velocity distribution in Fig.4a clearly show that short iPLA2 can form monomers and dimers in solution ...

It is unclear what reviewer refers to, since there are no monomeric species in the AUC experiment, corresponding to the WT protein. Monomer was only observed for W695E mutant designed to prove that the dimerization interface observed in the crystal structure, which is disrupted by this mutation, corresponds to the dimerization interface in solution. Importantly, the monomeric mutant is inactive.

..., but concentration-dependency (or rather the lack of it) in Fig.S4a raises more concerns.

In such experiment one will expect that increase in iPLA2 concentration will increase the probability of forming dimers. However, while more protein can be clearly detected at 0 and 0.1mM BS3 in 150nM iPLA2g6 samples, almost the same ratio between monomers and dimers can be seen at 1-10mM BS3 with both 50 and 150nM iPLA2 (it even looks like there may be more % of dimers in 50nM).

This assay was not designed as a quantitative experiment. The only purpose of the crosslinking experiment was to demonstrate the dimerization at low protein concentration and we presented results of two experiments performed under slightly different conditions. The ratio of monomers and dimers in gel is not equal to the ratio of monomers and dimers in solution. It is affected by multiple parameters such as the cross-linking efficiency, number of lysines close and far from the dimerization interface and the protein solubility in general. For example, the larger concentration of protein in the second experiment may require larger concentration of the crosslinker, which, under these conditions (>10 mM), can significantly affect protein solubility and crosslinking efficiency.

It is unclear if tetramers could be formed (the current blot is limited to 220kDa). To resolve these concerns, full range of MWs (covering potential tetramers) need to be presented in crosslinking experiments, and careful quantitative analysis of the ratio between monomers and dimers at different protein concentrations should be performed.

We did not observe tetramers in AUC in solution with high protein concentration and did not expect to find tetramers at low protein concentration. The full range of MWs is presented in cross-linking gel.

It will be also very important to demonstrate that short iPLA2 indeed comes as a dimer under physiological conditions in native gels. In the absence of such detailed analysis, the question remains, whether dimers seen in crystals are present and relevant for cellular function under physiological conditions.

AUC experiments are performed under physiologically-relevant conditions. Native gels, as suggested by the reviewer, are not an appropriate way to measure the absolute size of protein oligomers as the migration in the gel depends on multiple factors including size, shape, charge, dissociation constant and solution conditions compatible with electrophoresis. In contrast, AUC is a well-accepted gold standard for such measurements performed under equilibrium conditions in solution.

It is also worrisome that IQ domain was found to be buried within the CAT domain and inaccessible for CaM binding, which contradicts earlier studies that convincingly demonstrated the role of IQ in CaM binding and iPLA2 inhibition (some of which were published by several authors of the current study). So, there seem to be an internal contradiction, and either the earlier studies/results were incorrect (and may need retraction), or one can expect that the iPLA2 structure should accommodate accessibility of IQ for CaM binding/inhibition.

There is no contradiction. The old hypothesis is based on studies of isolated unstructured peptides tested outside of the protein structure. We also presented data of the isolated IQ motif interaction with CaM. This does not mean that it should interact with CaM when buried inside the folded globular structure. The conformation of the IQ motif observed in our crystal structure is not only a solid validated experimental result, but it can also be predicted with high confidence based on homologous structures of patatin and cPLA2. Homologous enzymes do have common structural folds and the position of the sequence corresponding to the IQ motif is well-defined in high resolution structures of related enzymes. This conformation was previously used to model the structure of the iPLA2b catalytic domain, as described in several earlier publications, and to successfully develop novel specific inhibitors, also published in several manuscripts.

(2) Another major concern is the continuous confusion between short and long splice variants and the lack of their clear distinction in this manuscript in terms of their potentially different structure, function, cellular localization and pathophysiology.

We clearly stated in the very first sentence of the Results section that the crystal structure and all solution studies are of a short variant. We did describe from the beginning where the insert place is located in the structure and discussed the potential role of the insert for the structure. We added additional references to publications that differentiate functional properties of two variants.

As an example, comparison of the structure of the short variant in current manuscript to the findings reported in {62} (Larsson et al, 1998), {81} (Engel et al, 2010), or {24} (Zhou et al, 2016) sounds

inappropriate, because all these (and some other) papers describe the properties of the full length 806-amino acid protein with a calculated molecular mass of 88 kDa, and not its short splice variant (752aa).

752 amino acids are identical in both variants. Therefore, the reported in our manuscript structure does relate to those publications.

In respect to the oligomeric form, in the revised manuscript we did discuss reasons for the potential overestimation of the molecular weight in published low resolution size exclusion chromatography experiments. In fact, the tetrameric form was also reported for the short variant. In our hands, the short variant also elutes in gel filtration as an apparent trimer or tetramer, depending on solution conditions and the gel filtration matrix. However, the same samples clearly represent a dimer when evaluated in AUC experiments, which is the well-established ultimate quantitative method to characterize protein molecular weight in solution.

In respect to the membrane localization, in a revised version we did discuss potential differences in properties of long variant due to the 54 residues-long insert.

Again, our structure of a naturally occurring in cell splice variant, represents 752 out of 806 residues of the long variant and this information should be extremely valuable for studies of both variants.

The authors failed to consider the fact that addition of 54-amino acid proline-rich insertion in the linker between ANK and CAT domains (residues 395–449) can significantly change their mutual orientation and dramatically affect the overall structural properties of the molecule, its cellular location and physiological function.

We did describe such a possibility in the text. However, it does not undermine the main conclusion of the paper, that the flexibility of the ANK domain permits its interaction with membrane proteins. With an additional insert in the long variant, the ANK domain is likely to be even more flexible. Importantly, these insert and ANK-related conformational changes will not affect major dimerization interface formed by CAT domains, which is the major finding of our structural, solution and mutagenesis studies.

Thus, all the claims and conclusions of the current study should be very clearly limited to the short splice variant. The lack of discrimination between specific variants can further fuel ongoing confusion regarding the cellular localization, physiological function and pathogenic dysfunction that can be different for different splice variants of iPLA2.

Again, we did discuss potential differences between the two variants in the text.

(3) Significant weakness of this manuscript is abundant references to iPLA2 role in different pathological conditions with apparent lack of explanation of the structure/functional roots for the differences between iPLA2 mutants association with distinct neurodegenerations. While a new figure 6 is appreciated, only few hand-picked mutations are shown, and revised manuscript does not provide sufficient explanation for the structural differences that could lead to association with INAD, PD or other conditions. For example, the current model does not seem to explain why R741 mutation to W leads to

the loss of catalytic activity and INAD, while its mutation to Q (R741Q) does not affect catalytic activity, and yet leads to PD. It is also unclear how other mutations can affect catalytic activity or other properties of this protein. Please, note that there is a mistake in the well-known PD-associated mutation in position 747: it is R747W, and not R747Q! Surprisingly, this mutation is not even mentioned in discussion, and it is totally unclear which aspect of iPLA2 function such mutation can affect, and how it can be linked to PD.

We have fixed the error.

A single structural snapshot is not sufficient to explain the potential effects of a constantly expanding list of disease mutations, particularly those with the residual catalytic activity. It will require a quantitative characterization of multiple biochemical properties, including not only the catalytic activity, but also all macromolecular interactions of iPLA2b. The structure, together with initial biochemical characterization of iPLA2b complex described in our paper, does provide an excellent framework to address the effect of each mutation in future.

The point of this section was to illustrate the importance of all domains for the protein function. As we stated, the mechanism of protein regulation and interactions remains a critical research topic. The current structure should provide enormous insights to advance such studies and the entire field.

A minor comment: labeling of genetic mutations using sequence of the shorter splice variant is a very unusual and confusing. To avoid confusion, disease-associated mutations are expected to represent the actual mutations within the gene sequence, and aa substitutions should be labeled based on the full length protein, not by aa numbers in the splice variants.

We have changed mutations labeling accordingly to the reviewer's suggestion.

(4) The claims for the ANK interactions with membrane proteins remain totally hypothetical (based on analogies to other proteins and generalized predictions, not on actual data), so, all such claims and speculations should be removed from Results and limited to Discussion.

This was removed from the Results and limited to the Discussion.

We also included new references where interaction of iPLA2b with membrane proteins was well-documented, as well as references to ankyrin repeats interactions with peptide motifs of membrane proteins.

Reviewer #3 (Remarks to the Author):

We fixed all points suggested by reviewer and we are very grateful for the help with improving the manuscript.

Thank you for the revised manuscript. It is much better written and more clear, and most issues have

been addressed satisfactorily. I would recommend publication after the following changes are made.

Line 29 in the abstract leads the reader to believe that the calmodulin was observed in the crystal structure or that there are conclusive data. It should say something less definitive, such as “the structure combined with biochemical evidence suggest that a single calmodulin is able to bind..”, or “we hypothesize that a single calmodulin is able to bind..”

Minor comments/edits:

Line 61 – edit this phrase “The AR is 33-amino acid long”

Line 91 – It would help to specify that the short variant is lacking the proline rich region and reference Fig. 1A here

Line 92 – write out SAD

Line 93 – Thank you for adding the homology with patatin, but that sentence reads a bit funny – the CAT domain is 32% homologous with patatin? The “four ARs of an ankryin-R protein” is something else but no homology is referenced? I would append this to the end of the previous sentence and change the period to a colon “: patatin, which is 32% homologous with the CAT domain, and four ARs of an ankryin-R protein”. However, I would also change homologous, which is vague, to similar or identical depending on which it is.

Line 101 – write out RMSD, change to read “an RMSD of 3.1 Å” (RMSD unit is not squared)

Line 281 – change to with the CAT domain

Line 387 – could you add the PDB codes for the proteins you used for molecular replacement along with the references, especially since four ARs of an ankryin-R proteins is vague.

Fig. 3 – uses both polyG and GGGVKG labeling, I would just pick one label

Reviewer # 2 apparently has no further comments.

Reviewers' comments:

Reviewer #3 (Remarks to the Author):

I still believe the science is worthy of publication, however, I also am quite frustrated with the authors argumentative and defensive tone in response to the reviews. While I agree with much of their point-by-point response to reviewer 1, specifically, it was difficult to follow/read, and again, argumentative.

While they indicated they made all the changes I asked for, the authors did not adjust their abstract as I requested in regards to a single calmodulin binding. This sentence in the abstract is misleading and there is no explanation for why it is unchanged.

Specifically in this version I would like to point out in line 145 - "confirming a potential interaction" - the wording is an oxymoron. Is it definitively confirmed as an interaction or does it support the hypothesis, etc?

Furthermore, the PDB validation report is significantly different from what the authors are reporting in table S1. I am well aware that the PDB validation report can have different values for the statistics, but they are usually just a little different. The completeness is 100(100) in table S1 for the native set and in the validation report it says they reported 61.5, which is more close to their SeMet data in the table. The I/sigma for the high res shell is 1.6 in their table S1 and much higher in the validation report (it says 4.94 at 4A). It should not be so inconsistent, especially for pre-refinement statistics. I am concerned they accidentally deposited a different version of the structure, or at least not the final data.

The authors should explain these discrepancies - I would want to see a final PDB report (with the PDB code listed) that matches more closely with the native data set they are reporting. Either their Table S1 needs to be modified or what is deposited needs to be modified. Also, can they add the CC1/2 for the high res shell to their table S1 (or is 0.7 the high res shell)?

Dear Reviewer,

We greatly appreciate your continuing support and patience and we apologize for missing your criticism about the overstatement of findings in the abstract. In the current version, we incorporated all your suggestions and answered all questions. The detailed point-by-point answers are below and the word file with highlighted changes is uploaded.

Reviewer #3 (Remarks to the Author):

I still believe the science is worthy of publication, however, I also am quite frustrated with the authors argumentative and defensive tone in response to the reviews. While I agree with much of their point-by-point response to reviewer 1, specifically, it was difficult to follow/read, and again, argumentative.

We appreciate your support and apologize for the lengthy, sometime, confusing arguments and defensive tone.

While they indicated they made all the changes I asked for, the authors did not adjust their abstract as I requested in regards to a single calmodulin binding. This sentence in the abstract is misleading and there is no explanation for why it is unchanged.

The sentence in abstract was changed to:

“The structure and solution studies suggest that a single calmodulin binds and allosterically inhibits both catalytic domains.”

Thank you very much for this important suggestion.

Several minor changes were introduced in abstract to keep it within the word limit.

Specifically in this version I would like to point out in line 145 - "confirming a potential interaction" - the wording is an oxymoron. Is it definitively confirmed as an interaction or does it support the hypothesis, etc?

It does support the hypothesis. We have modified the paragraph accordingly and included additional explanation.

Furthermore, the PDB validation report is significantly different from what the authors are reporting in table S1. I am well aware that the PDB validation report can have different values for the statistics, but they are usually just a little different. The completeness is 100(100) in table S1 for the native set and in the validation report it says they reported 61.5, which is more close to their SeMet data in the table. The $1/\sigma$ for the high res shell is 1.6 in their table S1 and much higher in the validation report (it says 4.94 at 4A). It should not be so inconsistent, especially for pre-refinement statistics. I am concerned they accidentally deposited a different version of the structure, or at least not the final data.

We carefully examined scaled data and final refinement statistics and found two issues, which affected the statistics in the table, but not the structure solution and the quality of refinement.

1) There is an apparent problem in HKL2000. The completeness in scalepack (HKL2000) output was 100% with 28492 reflections. However, there were only 24663 reflections in the output data file. We reported the problem to HKL Research Inc., but have not yet received a response. The problem was reproducible with a different version of HKL in a different facility. It appears that “autocorrection” option led to rejection of certain reflections due to anisotropic diffraction, but the completeness and all statistics in the log file was calculated for all processed reflections before rejection. Indeed, scaling without “autocorrection” resulted in a similar statistics, while the processing of these data (not subjected to “autocorrection”) with STARANISO server (Global Phasing Limited) resulted in large number of rejections (25934 reflections after rejection) similar to that of in “autocorrection” HKL2000 output. This led to underestimation of some statistical indicators (e.g. I/σ 1.6 versus 4.2) and overestimation of the completeness. We believe that the initial statistics of HKL2000 scaling should be reported in the table, as it reflects the data scaling process. To report anisotropy related rejections, we included in the table additional statistics parameters reported by Xtriage program implemented in Phenix suit.

Importantly, this additional statistics better reflects true resolution limits of the anisotropic diffraction, which we verified with STARANISO and UCLA Diffraction Anisotropy servers. Overall, maps calculated with data sets produced by HKL2000 with “autocorrection” option were still more detailed and of a better quality, then those generated by other anisotropy servers (see examples below), and these data sets was used for phasing and refinement.

2) The refinement data set unintentionally included anomalous data, because the output from HKL2000 scaling with “autocorrection” was in anomalous format (even though “anomalous differences” were very small and few) and the Free-R data set was also created for SeMet data in anomalous format, once we obtained first interpretable electron density map. In preparation of the PDB report the anomalous pairs were treated as independent reflections leading to the discrepancy in the number of reflections. We have now properly averaged and merged data. Although anomalous differences were negligible and number of anomalous pair was small, we performed several cycles of additional refinement to verify consistency in refinement statistics. Data and coordinates were updated in PDB deposition, accordingly. A validation report from new deposition is attached.

These problems did not affect the quality of the final structure. We used the electron density maps as overriding quality control.

The authors should explain these discrepancies - I would want to see a final PDB report (with the PDB code listed) that matches more closely with the native data set they are reporting. Either their Table S1 needs to be modified or what is deposited needs to be modified. Also, can they add the CC1/2 for the high res shell to their table S1 (or is 0.7 the high res shell)?

Yes, 0.7 is CC1/2 for highest resolution shell. We added comment to the table notes.

Overall, the “true” resolution of diffraction data is subject to interpretation. Due to anisotropy, overall completeness of the final data beyond 4.4 Å is low, but reflections along c^* axis are much stronger up to 3.95 Å (e.g. discrepancy in I/σ value). We followed the principle of including all valuable reflections, judged by the quality of the electron density map.

Examples of 2Fo-Fc electron density maps calculated using data scaled without and with “autocorrection” during earlier stages of refinement.

- 1) Scaled without “autocorrection” and run through the anisotropy analysis server
<https://services.mbi.ucla.edu/anisocscale/>

Res: 3.95A

Res: 4.6A

- 2) Scaled with “autocorrection” option (3.95 A)